

# Relative importance of gas uptake on aerosol and ground surfaces characterized by equivalent uptake coefficients

Meng Li[1], Hang Su[2,1], Guo Li[1], Nan Ma[2], Ulrich Pöschl[1], and Yafang Cheng[1]

[1] Max Planck Institute for Chemistry, Mainz, 55118, Germany
5 [2] Center for Air Pollution and Climate Change Research (APCC), Institute for Environmental and Climate Research (ECI), Jinan University, Guangzhou, 511443, China

*Correspondence to*: Hang Su (h.su@mpic.de) and Yafang Cheng (yafang.cheng@mpic.de)

**Abstract.** Quantifying the relative importance of gas uptake on the ground and aerosol surfaces helps to determine which processes should be included in atmospheric chemistry models. Gas uptake by aerosols is often characterized by an effective 10 uptake coefficient ($\gamma_{eff}$), whereas gas uptake on the ground is usually described by a deposition velocity ($V_d$). For efficient comparison, we introduce an equivalent uptake coefficient ($\gamma_{eqv}$) at which the uptake flux of aerosols would equal that on the ground surface. If $\gamma_{eff}$ is similar to or larger than $\gamma_{eqv}$, aerosol uptake is important and should be included in atmospheric models. In this study, we compare uptake fluxes in the planetary boundary layer (PBL) for different reactive trace gases ($O_3$, $NO_2$, $SO_2$, $N_2O_5$, $HNO_3$, $H_2O_2$), aerosol types (mineral dust, soot, organic aerosol, sea salt aerosol), environments (urban, 15 agricultural land, Amazon forest, water body), seasons, and mixing heights.

For all investigated gases, $\gamma_{eqv}$ ranges from $10^{-6} \sim 10^{-4}$ in polluted urban environments to $10^{-4} \sim 10^{-1}$ under pristine forest conditions. In urban areas, aerosol uptake is relevant for all species ($\gamma_{eff} \geq \gamma_{eqv}$) and should be considered in models. On the contrary, contributions of aerosol uptakes in Amazon forest are minor compared to the dry deposition. Phase state of aerosols could be one of the crucial factors influencing the uptake rates. Current models tend to underestimate the $O_3$ uptake on liquid 20 organic aerosols which can be important especially over regions with $\gamma_{eff} \geq \gamma_{eqv}$. $H_2O_2$ uptakes on a variety of aerosols is yet to be measured at laboratory and evaluated.

Given the fact that most models have considered their uptakes on the ground surface, we suggest also considering the $N_2O_5$ uptake by all types of aerosols, $HNO_3$ and $H_2O_2$ uptakes by mineral dust, $O_3$ uptake by liquid organic aerosols and $NO_2$, $SO_2$, $HNO_3$ uptakes by sea salt aerosols in atmospheric models.

25 **1 Introduction**

Multiphase processes play an essential role in atmospheric chemistry and atmosphere-biosphere exchange (Ravishankara, 1997; Ammann et al., 1998; Gard et al., 1998; Usher et al., 2003; Bauer et al., 2004; Fowler et al., 2009; Kolb et al., 2010; Su et al., 2011, 2013; Herrmann, 2003, 2015; Ammann et al., 2013; Oswald et al. 2013; George et al., 2015; McNeill, 2015;





Pöschl and Shiraiwa, 2015; Quinn et al., 2015; Cheng et al., 2016; Froehlich-Nowoisky et al. 2016; Lappalainen et al. 2016; Tang et al., 2016; Meusel et al. 2018). It not only affects the atmospheric trace gases concentrations but also modifies the properties of condensed phases, commonly known as the aging process (Song and Carmichael, 1999; Cheng et al., 2006, 2012; Rudich et al., 2007; Andreae 2009; Jimenez, et al., 2009; Gunthe et al., 2011; Ditas et al., 2018). In the planetary

boundary layer, aerosols and ground provide two kinds of surfaces for multiphase reactions. In previous gas uptake studies, different formulations have been used to describe and parameterize the gas uptake processes (Wesely, 1989; Ravishankara, 1997; Jacob, 2000; Wesely and Hicks, 2000; Zhang et al., 2003; Ammann and Pöschl, 2007; Pöschl et al., 2007; Wesely, 2007).

A variety of ground surfaces, including vegetation, water, rock, road etc., can take up gaseous species through dry deposition,
thus having significant impacts on the budget of these reactive gases and on the physicochemical properties of the ground surface itself (Lelieveld and Dentener, 2000; Ashmore, 2005). Dry deposition is one of the major removal pathways for most gaseous species and has been extensively parameterized in atmospheric models (Wesely and Hicks, 2000; Zhang et al., 2002, 2003). A resistance model, which consists of the aerodynamic resistance, quasi-laminar resistance and surface resistance, has been widely applied to calculate the dry deposition flux in global and regional atmospheric models (see Fig. 1, Wesely and
Hicks, 2000; Wesely, 2007). The dry deposition velocity, $V_d$ (in unit of cm s$^{-1}$) calculated as the reciprocal of the total resistance, is the key parameter to describe the uptake fluxes on the ground.

From late 1990s, the importance of reactive uptake of gases by aerosols has been commonly accepted (Ravishankara, 1997; Gard et al., 1998; Jacob, 2000). Gas uptake by aerosols not only influences the fate of reactive gases, but also changes the physio-chemical properties of atmospheric aerosols (Kolb et al., 2010). Taking account of the multiphase chemistry is
proven a key factor to explain the observations and improve the model performances (Zhang and Carmichael, 1999; Song and Carmichael, 2001; Liao and Seinfeld, 2005; Wang et al., 2006; McNaughton et al., 2009; Wang X et al., 2012; Zheng B et al., 2015; Chen et al. 2018; Mu et al., 2018). Compared to dry deposition, the parameterization of gas uptake on aerosols is more challenging (Jacob, 2000; Pöschl and Shiraiwa, 2015). The mass transfer between gases and aerosols can be described by the resistance model in analogy to electrical circuit which decoupled the physio-chemical limitations in the gas phase,
gas-surface interface and the bulk phase under (quasi-) steady state conditions (Schwartz and Freiberg, 1981; Schwartz, 1986; Kolb et al., 1995). A simplified scheme, which relies on the formulation of effective uptake coefficient ($\gamma_{eff}$) has been widely used in current atmospheric models (Jacob, 2000; Liao and Seinfeld, 2005; Wang K et al., 2012). Growing numbers of laboratory studies have reported $\gamma_{eff}$ for various trace gases and aerosol particles that are potentially important for atmospheric chemistry, such as $O_3$, $NO_2$, $SO_2$, $N_2O_5$, $HNO_3$ on mineral dust (Ullerstam et al., 2002; Mogili et al., 2006;
Vlasenko et al., 2006; Wagner et al., 2008; Ndour et al., 2009), soot (Rogaski et al., 1997; Longfellow et al., 2000; Al-Abadleh and Grassian, 2000; Saathoff et al., 2001), and sea salt aerosols (Mochida et al., 2000; Gebel and Finlayson-Pitts, 2000; Hoffman et al., 2003; Thornton and Abbatt, 2005; Ye et al., 2010). A series of evaluations on the kinetic and



photochemical data for the multiphase reactions were conducted afterwards (Crowley et al., 2010, 2013; Ammann et al., 2013; Burkholder et al., 2015). Pöschl et al. (2007) and the follow-up studies (e.g., Shiraiwa et al., 2010, 2011) developed a comprehensive kinetic model framework, enabling consistent and unambiguous descriptions of mass transfer and chemical reactions in aerosol systems.

However, which kind of surfaces is more important for gas uptake in the planetary boundary layer (PBL)? The answer is not straightforward because of the following reasons:

(1) though the surface of the Earth seems to be much larger than that of tiny aerosols, its contribution is diluted by the large volume of the PBL, resulting in a surface to volume ratio close to that of aerosol. For example, for a PBL height of 1000 m, the corresponding surface to volume ratio is 1000 $\mu m^2$ $cm^{-3}$, comparable to aerosol surface area concentrations of 200 ~ 2000

$\mu m^2$ $cm^{-3}$ for urban areas (Woo et al., 2001; Stanier et al., 2004; Wu et al., 2008, 2017; Ma and Birmili, 2015), and 200 ~ 1000 $\mu m^2$ $cm^{-3}$ for rural environments (Ma et al., 2014; Ma and Birmili, 2015; Wu et al., 2017; Held et al., 2008).

(2) different formulations also hinder the comparison. As illustrated above, different schemes, formulations and terminologies are applied to calculate the uptake fluxes on ground and aerosols. The dry deposition velocity ($V_d$) is the fundamental parameter to describe the deposition process on the ground while the effective uptake coefficient ($\gamma_{eff}$) is used to

describe the uptake fluxes on aerosols.

In this study, we conducted a comparative assessment of the gas uptake on both ground and aerosol surfaces. Our goal is to identify the prevailing multiphase process in the PBL, and especially those processes that have not yet been sufficiently considered in atmospheric models. Section 2 described the methods of calculation and comparison. We presented and discussed the main results in Section 3, which is followed by a summary of our major findings in Section 4.

**2 Methods**

In this work, we compared the relative importance of gas uptake by the ground and aerosols based on their uptake fluxes. In this comparison, resistance models were applied to calculate uptake fluxes on both ground and aerosol surfaces (see Fig. 1) as detailed below. The uptake fluxes of six reactive gases ($O_3$, $NO_2$, $SO_2$, $N_2O_5$, $HNO_3$, $H_2O_2$) were calculated and compared for four typical land use categories (urban, agricultural land, Amazon forest, water) and five aerosol types (mineral dust, soot,

solid organic aerosol, liquid organic aerosol, sea salt aerosol). These species were chosen considering their potential importance regarding dry deposition on the ground and uptake on aerosols within the troposphere.





## 2.1 Ground gas uptake

Dry deposition fluxes were calculated following the scheme and parameters of Wesely (1989) and Zhang et al. (2003). As shown in Fig. 1, the resistance model applied to characterize the dry deposition process includes the aerodynamic resistance ($R_a$), quasi-laminar resistance ($R_b$) and surface resistance ($R_c$). The basic equations for the flux calculations are:

$$F_{grd} = -V_d[X_g] \times 10^{-2} \tag{1}$$

$$V_d = \frac{1}{R_{grd}} = \frac{1}{R_a + R_b + R_c} \tag{2}$$

where $F_{grd}$ represents the gas deposition fluxes on various ground surfaces (mol m$^{-2}$ s$^{-1}$); $V_d$ represents the deposition velocity (cm s$^{-1}$); $[X_g]$ is the averaged gas concentration (mol m$^{-3}$); $R_{grd}$ is the total resistance in the dry deposition process (s cm$^{-1}$), composed of $R_a$, $R_b$ and $R_c$. The detailed equations and parameterization scheme for determination of $R_a$, $R_b$ and $R_c$ are provided in the supplement. A neutral meteorological condition was assumed in the calculation. We present the key input parameters and the calculated $V_d$ in Table S1 and Table S2, respectively.

## 2.2 Aerosol gas uptake and the effective uptake coefficient ($\gamma_{eff}$)

The net flux of gas $X$ from gas phase to the condensed phase ($J_{net}$, mol m$^{-2}$ s$^{-1}$) for one aerosol particle can be expressed as Eq. (3) under (quasi-) steady-state conditions (Pöschl et al., 2007):

$$J_{net} = \frac{\omega \gamma_{eff}}{4}[X_g] \tag{3}$$

The effective uptake coefficient, $\gamma_{eff}$, represents the number of gas molecules taken by the aerosol particle divided by the number of those impacting onto the particle surface (Pöschl et al., 2007); $\omega$ is the mean thermal velocity (m s$^{-1}$), we use a typical value of 300 m s$^{-1}$ in this study; $[X_g]$ is the averaged gas concentration far away from the aerosol surface (mol m$^{-3}$).

$$\frac{1}{\gamma_{eff}} = \frac{1}{\Gamma_g} + \frac{1}{\alpha} + \frac{1}{\Gamma_b} \tag{4}$$

As shown in Fig. 1, resistance models have been widely applied to quantify the mass transfer of gases to aerosol particles. For gas uptake on liquid droplets, following the resistance model as described by Eq. (4), the overall resistance $1/\gamma_{eff}$ is composed of three resistor terms due to gas diffusion ($1/\Gamma_g$), interfacial mass transfer ($1/\alpha$) and bulk diffusion and reaction ($1/\Gamma_b$) (Pöschl et al., 2007). The conductance of gas diffusion is commonly calculated based on $\Gamma_g = 8D_g\omega^{-1}d_p^{-1}$, where $D_g$ is the diffusion coefficient of gas $X$ in the gas phase (m$^2$ s$^{-1}$), and $d_p$ represents the aerosol particle diameter. For atmospheric aerosols with a diameter of ~0.2 μm, the related gaseous mass accommodation tends to be collision and uptake limited (Jacob, 2000), thus we neglect the diffusion resistance in the gas phase in the following analyses and discussions.

Given a mixing height of $h$, and aerosol surface area density of $A$ (particle surface area per unit volume of air, μm$^2$ cm$^{-3}$), the total uptake flux of gas $X$ by aerosols ($F_{aer}$, mol m$^{-2}$ s$^{-1}$) is:

$$F_{aer} = J_{net}Ah = \frac{\omega \gamma_{eff}}{4}Ah[X_g] \times 10^{-6} \tag{5}$$



where $10^{-6}$ is the unit conversion factor. We summarized the measured uptake coefficients for a variety of gas species and aerosol types at both initial state and steady state in Table 1 (details in Table A.1 ~ A.4). They are mainly derived from the measured values in literatures, reviewed data of IUPAC (International Union of Pure and Applied Chemistry) Task Group on Atmospheric Chemical Kinetic Data Evaluation (Crowley et al., 2010, 2013; Ammann et al., 2013; available at

http://iupac.pole-ether.fr/), and NASA-JPL (Jet Propulsion Laboratory, Burkholder et al., 2015) (see references in Table A.1 ~ A.4). As we focus on PBL, those $\gamma_{eff}$ measured at room temperatures (~298K) are mainly presented. Gas uptakes at very low temperature (e.g., polar region, stratosphere) are out of scope of this study and should be explored in future work.

Though the initial and steady-state uptake coefficients are listed, it should be noted that the values at initial state may not be appropriate for direct application in chemical transport models (CTMs) considering the subsequent surface saturation and

depletion of reactants for several cases (e.g., on mineral dust and soot, Ndour et al., 2009; Stephens et al., 1986; Ammann et al. 1998; Kalberer et al. 1999). In general, the upper limit and lower limit are determined based on those derived using the geometric surface and the BET (Brunauer-Emmett-Teller) surface, respectively. Preferences are given to those measured at steady state using ambient aerosols, or recommended values by the IUPAC group with relatively high reliability. As shown in Table A.1, more than 3 orders of magnitude of variances are found for $SO_2$ and $O_3$ uptake on mineral dust depending on

the gas concentration and aerosol components (Michel et al., 2002, 2003; Mogili et al., 2006; Ullerstam et al., 2002, 2003; Li et al., 2006). Large discrepancies also exist for $SO_2$ and $HNO_3$ uptake on soot (Longfellow et al., 2000; Saathoff et al., 2001; Xu et al., 2015). For $H_2O_2$, limited measurements of $\gamma_{eff}$ have been conducted for aerosols apart from mineral dust and soot.

## 2.3 Uptake coefficient at equivalent flux ($\gamma_{eqv}$)

To help the evaluation, we define an uptake coefficient at equivalent flux $\gamma_{eqv}$. Here, $\gamma_{eqv}$ is the effective uptake coefficient on

aerosols when the ground flux equals the aerosol flux within the PBL. When $\gamma_{eff} > \gamma_{eqv}$, the aerosol surfaces are more important than the ground surfaces regarding the gas uptake and vice versa. By letting $F_{grd}$ equal $F_{aer}$, we can derive the expression of $\gamma_{eqv}$ as below.

$$\gamma_{eqv} = \frac{4}{3} \frac{V_d}{Ah} \times 10^2 \tag{6}$$

and in a typical mixing height of 300m, we have

$$\gamma_{eqv} = \frac{V_d}{2.25A} \tag{7}$$

According to Eq. (6), $\gamma_{eqv}$ is proportional to $V_d$, and is inversely proportional to aerosol surface area density and the mixing height. We calculated a series of $\gamma_{eqv}$ for a variety of gas species, land use categories, seasons, aerosol surface area densities ($A$) and mixing heights ($h$).

As defined, $\gamma_{eqv}$ reflects the relative importance of gas uptake on aerosols compared to those on the ground surfaces. Larger

$\gamma_{eqv}$ indicates higher probability for gases to deposit on the ground rather than on aerosols for further chemical reactions on





surface and bulk, and vice versa. Low dry deposition velocities and high loadings of aerosols providing large amounts of surface reaction sites can benefit gas uptake on aerosols. The derived $\gamma_{eqv}$ and $\gamma_{eff}$ from laboratory measurements are compared in Sect. 3.

## 3. Results and discussion

To estimate the possible range of $\gamma_{eqv}$ for different environments, we designed different scenarios with mixing height $h$ varying between 100 m and 1.0 km (a typical value of 300m), and $A$ varying with land use categories as follows:

(a) Range of $A$. We set the range of $A$ based on measurements in various environments collected in literature. $A$ are in the range of 200 ~ 2000 $\mu m^2$ $cm^{-3}$ for aerosols in the urban area (Woo et al., 2001; Stanier et al., 2004; Wu et al., 2008, 2017; Ma and Birmili, 2015), 200 ~ 1000 $\mu m^2$ $cm^{-3}$ in agricultural land (sub-urban and rural, Held et al., 2008; Su et al. 2008; Ma

et al., 2014; Ma and Birmili, 2015; Wu et al., 2017), 8 ~ 700 $\mu m^2$ $cm^{-3}$ in Amazon forest (Zhou et al., 2002; Rissler et al., 2006; Pöschl et al., 2010; Andreae et al. 2015), and 20 ~ 200 $\mu m^2$ $cm^{-3}$ for sea salt aerosols (SSA, O'Dowd et al., 1997; Ghan et al., 1998; Lewis and Schwartz, 2004).

(b) Typical $A$ (corresponding to the typical $\gamma_{eqv}$ in Fig. 3 ~ Fig. 5). We use 1050 $\mu m^2$ $cm^{-3}$ for the urban environment (Wang et al., 2017), 230 $\mu m^2$ $cm^{-3}$ for the agricultural land (Held et al., 2008), 46 $\mu m^2$ $cm^{-3}$ for the Amazon forest (Rissler et al.,

2006), and 76 $\mu m^2$ $cm^{-3}$ for SSA (canonical distribution at wind speed of 10 m $s^{-1}$, Lewis and Schwartz, 2004).

It should be noted that the above ranges and the typical values of $A$ are derived from current available experiments to support our analyses and discussions in this study, but still cannot cover all cases of particle distributions in the world.

Figure 2 shows the calculated $\gamma_{eqv}$ over a range of dry deposition velocity and aerosol surface area densities at a mixing height of 300m. $V_d$ for different scenarios were calculated based on the resistance scheme illustrated above, showing a range

of 0.01 ~ 2.3 cm $s^{-1}$, with lowest for $NO_2$ and highest for $N_2O_5$ and $HNO_3$ (details in Table S2). Aerosol surface area densities covered a range of 8.6 $\mu m^2$ $cm^{-3}$ to 2139 $\mu m^2$ $cm^{-3}$, from pristine rainforest to polluted megacities. We show the calculated $\gamma_{eqv}$ at typical conditions (typical $A$ as described above, $h$=300m) by season in Table S3 and detailed illustrated $\gamma_{eqv}$ for each gas species in sections below. As shown in Fig.2, $\gamma_{eqv}$ decreases with increase of $A$, which is closely related to the air pollution level, and increases with increasing $V_d$.

For small $V_d$ ( ≤ 0.1 cm $s^{-1}$), $\gamma_{eqv}$ lie in the range of $10^{-5}$ ~ $10^{-4}$ for clean regions, such as Leipzig, Melpitz, Pittsburgh, and reduced to $10^{-6}$ ~ $10^{-5}$ under polluted cities including Beijing and Wangdu. This low dry deposition can be found for $NO_2$ above the urban ground (0.03 cm $s^{-1}$, seasonal mean), and $O_3$, $NO_2$, $SO_2$ and $H_2O_2$ on water body (0.07 cm $s^{-1}$, 0.01 cm $s^{-1}$, 0.03 cm $s^{-1}$, and 0.08 cm $s^{-1}$, respectively). The downward shift of $\gamma_{eqv}$ with larger aerosol surface area density suggests an increasing importance of gas uptake in polluted areas than clean areas.





With the increase of $V_d$ (> 0.1 cm s$^{-1}$), $\gamma_{eqv}$ increases to $10^{-5} \sim 10^{-2}$ accordingly. In pristine region of Amazon forest, $\gamma_{eqv}$ can reach up to $10^{-2}$. The lowest $\gamma_{eqv}$ is $2.1 \times 10^{-5}$ during haze events with high concentrations of fine particulate matter and surface area in the PBL ($A$=2139 $\mu m^2$ cm$^{-3}$). In this study, this range of $V_d$ covers most of the investigated cases, including $O_3$, $SO_2$, $H_2O_2$ on urban/Amazon forest/agricultural land, $NO_2$ on agricultural land/Amazon forest, and $N_2O_5$, $HNO_3$ on all land

use types (see Table S2). Thus we can derive a general criterion of $\gamma_{eff} > 10^{-5}$ conservatively for aerosol uptake to compete with the dry deposition.

In the following, we further compared $\gamma_{eqv}$ to the laboratory measurements of $\gamma_{eff}$ for different reactive gases ($O_3$, $NO_2$, $SO_2$, $N_2O_5$, $HNO_3$, $H_2O_2$). The uptake coefficients at initial state are in general 1~3 magnitudes higher than those at steady-state (see Table 1 and Fig. 3~5). Considering the timescale of gas uptake by aerosols in the real world and applications in models,

we mainly focused on the comparisons of $\gamma_{eqv}$ and the steady-state $\gamma_{eff}$ in the following discussions.

### 3.1 $O_3$

Under typical conditions (typical $A$ by land use, $h$=300m, as illustrated above), $\gamma_{eqv}$ for $O_3$ are determined between $9.2 \times 10^{-5}$ and $2.2 \times 10^{-3}$, lowest in urban and highest in the Amazon forest. The extended range is $1.4 \times 10^{-5} \sim 3.8 \times 10^{-2}$, varying with particle area densities and mixing heights (Fig. 3). There are overlaps between $\gamma_{eqv}$ and $\gamma_{eff}$ for liquid organic aerosols among

all investigated typical environments, and other types of aerosols under favorable circumstances for aerosol uptake in urban. $\gamma_{eff}$ lie below $\gamma_{eqv}$ for other combinations of aerosol types and land use categories.

We can only expect comparable uptake between ground and aerosol surfaces of mineral dust, soot, solid organic aerosol, and SSA at high aerosol loadings in urban (e.g., $A$=1400 $\mu m^2$ m$^{-3}$, Beijing) and/or high mixing layers (e.g., $h$=1.0 km). Combined with the measured uptake coefficients which lie in the range of $1.0 \times 10^{-7}$ to $1.6 \times 10^{-4}$ for soot, $1.1 \times 10^{-5}$ to $3.0 \times 10^{-3}$ for

liquid organic aerosols and $1.3 \times 10^{-6}$ to $1.0 \times 10^{-4}$ for SSA, we can expect high uptake fluxes of $O_3$ on these three kinds of aerosols when corresponding $\gamma_{eff}$ larger than $10^{-4}$ for other ground surfaces.

Complexity comes from the organic aerosols, of which the phase state has a large impact on the uptake and is subject to the temperature, relative humidity and particle size (see Fig. 3) (Virtanen et al. 2010; Cheng et al. 2015). For liquid organic aerosols, the measured $\gamma_{eff}$ show large variances from $10^{-5}$ to $10^{-3}$ and corresponding $\gamma_{eqv}$ fall into this range, demonstrating

that $O_3$ uptake on aerosols is comparable to that on the ground. Thus, multiphase reactions of $O_3$ on liquid organic aerosols should be included in atmospheric models. This is also consistent with the findings of Mu et al. (2018), which demonstrates the importance of the phase state of aerosols in multiphase reactions and transport of polycyclic aromatic hydrocarbons to improve the model performances at both regional and global scales.

Shiraiwa et al. (2017) shows the global map of SOA (secondary organic aerosol) phase state at the earth's surface. SOA in

Southern China, Amazon forest and South Africa are mainly in liquid phase within PBL. For these regions, the comparable



uptake fluxes for $O_3$ on liquid organic aerosols compared to the dry deposition demonstrate the importance of aerosol uptake. Dry deposition is one of the major sinking pathways for $O_3$ (Ganzeveld and Lelieveld, 1995). The uptakes of $O_3$ by aerosols are expected to contribute comparable sink fluxes as dry deposition regionally. Inclusion of the $O_3$ uptake by organic aerosols in these regions will increase the deposition rate of $O_3$ on aerosols, affect its lifetime, and further affect the fate of

$HO_x$, $NO_x$ through chemical reactions in the gas phase.

## 3.2 $NO_2$

For $NO_2$, $\gamma_{eqv}$ are generally above the upper limit of $\gamma_{eff}$ in urban, agricultural land and forest environments, as shown in Figure 4, demonstrating that the ground surfaces are of greater importance than aerosols. Overlaps are found for SSA on various land use types and also for liquid organic aerosols under the urban environment.

$NO_2$ tend to deposit on ground surface instead of on mineral dust particles, soot and solid organic aerosols. As reviewed in Table A.1~A.3, the effective uptake coefficient of $NO_2$ on these three kinds of aerosols are at magnitudes of $< 10^{-6}$ under steady-state conditions. For $A$ ranging from 46 $\mu m^2\ cm^{-3}$ (Amazon) to 1050 $\mu m^2\ cm^{-3}$ (Wangdu) and mixing height of 300 m, $\gamma_{eqv}$ of $NO_2$ lie between $1.4 \times 10^{-5}$ and $1.3 \times 10^{-3}$, 1~3 orders of magnitudes larger than $\gamma_{eff}$ on these three kinds of aerosols. Increasing the PBL mixing height and aerosol surface area may reduce $\gamma_{eqv}$ by ~1-2 magnitudes, but are still above the

measured $\gamma_{eff}$ at steady state.

The reactive uptake coefficients of $NO_2$ by SSA were quantified in the range of $10^{-6}$ to $10^{-4}$, demonstrating the ability of ambient sea salt aerosols to take in chemical species like $NO_2$ (Harrison and Collins, 1998; Yabushita et al., 2009; Ye et al., 2010). The high uptake coefficients observed for SSA ($6.0 \times 10^{-7} – 3.0 \times 10^{-4}$) are probably attributed to the reactions of $Cl^-$ with dissolved $NO_2$ in aqueous phase (Msibi et al., 1993; Harrison and Collins, 1998; Yabushita et al., 2009). The

overlapped values of $\gamma_{eqv}$ and $\gamma_{eff}$, show that the $NO_2$ uptake by SSA is comparable to the uptake by land surface or water body in coastal areas and therefore should be taken into account in atmospheric models.

Another important process is the $NO_2$ uptake on liquid organic aerosols ($\gamma_{eff}$ in the range of $2.2 \times 10^{-7} – 7.0 \times 10^{-6}$) in urban area of high $A$. As shown in Fig. 4, the lower limit of $\gamma_{eqv}$ in urban is $\sim 2.2 \times 10^{-6}$, lying in the range of $\gamma_{eff}$. The uptake coefficients of $NO_2$ on pure water are estimated around $10^{-7} \sim 10^{-6}$ driven by low solubility and slow hydrolysis rates

(Kleffmann et al., 1998; Gutzwiller et al., 2002; Ammann et al., 2005; Komiyama and Inoue, 1980). Harrison and Collins (1998) reported a high $\gamma_{eff}$ of $5.4 \sim 5.8 \times 10^{-4}$ for $NO_2$ uptake on ammonium sulfate aerosols at high RH (RH=50%, 85%). Presence of reactants such as inorganics of $HSO_3^-$ or phenolic compounds in aqueous aerosols can promote the uptake significantly through chemical reactions with dissolved $NO_2$ to $10^{-5} \sim 10^{-4}$ (Msibi et al., 1993; Lee and Tang, 1998; Spindler et al. 2003; Ammann et al., 2005; Yabushita et al., 2009; Su et al. 2008; Cheng et al., 2016). Multiple measurements and

modeling work have also pointed out that high alkalinity of aqueous aerosols is key to promote the reactions and further





increase the $NO_2$ uptake rates (Ammann et al., 2005; Herrmann et al., 2015; Cheng et al., 2016). Therefore, the $NO_2$ uptake on alkaline aqueous aerosols containing organic/inorganic reactants is competitive in the urban atmosphere, and should be detailed addressed in models. In Amazon forest, where $A$ is too low (46 $\mu m^2$ $cm^{-3}$), corresponding to a $\gamma_{eqv}$ on the order of $10^{-3}$, even a high $\gamma_{eff}$ of $10^{-4}$ is not sufficient to compete with the uptake by the ground surfaces.

In summary, the $NO_2$ uptake coefficients on liquid aerosol droplets can vary by three orders of magnitude with aerosol compositions ($10^{-7}$~$10^{-4}$). On liquid organic aerosols and sea salt aerosols, the uptake can reach up to $10^{-6}$~$10^{-4}$ through chemical reactions (Abbatt and Waschewsky, 1998; Ammann et al., 2005; Yabushita et al.,2009), significantly larger than the uptake on pure water of $10^{-7}$~$10^{-6}$ (Lee and Tang, 1988; Kleffmann et al., 1998; Gutzwiller et al., 2002). For liquid ammonium sulfate aerosols, discrepancies with two orders of magnitude ($10^{-6}$~$10^{-4}$) in $\gamma_{eff}$ are found with reasons

unexplained yet (Harrison and Collins, 1998; Tan et al., 2016). Considering these variances, aerosol components are important to parameterize the $\gamma_{eff}$ in atmospheric models.

**3.3 $SO_2$**

The calculated $\gamma_{eqv}$ of $SO_2$ vary between $1.0 \times 10^{-4}$ and $2.1 \times 10^{-3}$ for land surfaces, and $1.7 \times 10^{-4}$ above water body under typical conditions. As shown in Fig. 4, the $SO_2$ uptake by mineral dust is comparable to the ground uptake in urban, and

under favorable conditions over agricultural land and water body. For soot, aerosol uptake is magnitudes lower than those on the ground ($\gamma_{eqv} \geq \gamma_{eff}$), thus is unimportant for $SO_2$. For SSA, $\gamma_{eff}$ of $3.2 \times 10^{-3}$ ~ $1.7 \times 10^{-2}$ has been reported for $SO_2$ at aerosol pH of 5.4~6.6, which is high enough to compete with dry depositions over most environments (Gebel et al., 2000). Additional reactions of $SO_2$ and $O_3$ in alkaline solutions are found to promote the $SO_2$ uptake and form sulfate on SSA at first stage (Laskin et al., 2003). However, aerosol acidification due to production of H+ has been suggest to quickly suppress

the oxidation process in the real world (Alexander et al., 2005). We suggest including both the $SO_2$ uptake on SSA and the aerosol acidification process in models.

The extended range of $\gamma_{eqv}$ is $1.6 \times 10^{-5}$~$1.6 \times 10^{-3}$, $5.5 \times 10^{-5}$~$2.8 \times 10^{-3}$, and $1.9 \times 10^{-5}$~$1.9 \times 10^{-3}$ for urban, agricultural land and water body, respectively. $\gamma_{eff}$ of mineral dust falls in this range under high aerosol loadings or high mixing heights. The wide range of $\gamma_{eff}$ for mineral dust ($1.5 \times 10^{-8}$ to $6.3 \times 10^{-4}$) is a big challenge regarding its application in models, because it can be

affected by the presence of oxidant, phase state, components of the tested dust and the use of surface area in calculation (Huss et al., 1982; Ullerstam et al., 2003; Li et al., 2006; Alexander et al., 2009; Zhang et al., 2018). We further discuss the $SO_2$ uptake on mineral dust by different conditions as below.

Under dry conditions (as reviewed in Table A.1), $\gamma_{eff}$ are measured on the order of $10^{-7}$ ~ $10^{-4}$ (Goodman et al., 2001; Usher et al., 2002; Ullerstam et al., 2003; Adams et al., 2005; Li et al., 2006). IUPAC recommended an averaged value of $4 \times 10^{-5}$ for

atmospheric modeling, based on measurements using airborne aerosols (Usher et al., 2002; Adams et al., 2005).



In environments with high RH, water can enhance or inhibit the uptake by affecting reactive sites, varying with experimental conditions (Huang et al., 2015; Zhang et al., 2018). On the other hand, the uptake rate can be improved by several factors and/or aqueous chemical reactions, such as presence of $O_3$, $H_2O_2$, and transition metal ions (TMIs), which strongly depends on the aerosol pH (Jayne et al., 1990; Li et al., 2006; Cheng et al., 2016; Zhang et al., 2018). The initial $\gamma_{eff}$ of $SO_2$ on pure

water can reach as high as $10^{-3} \sim 0.1$ varying with pH (Gardner et al., 1987; Worsnop et al., 1989; Jayne et al., 1990; Ponche et al., 1993). Depending on aerosol pH and oxidant concentrations, the regimes of $SO_2$ uptake and sulfate formation may transit from TMI- or $H_2O_2$-dominated regime to $NO_2$- or $O_3$-dominated regime (Cheng et al., 2016). In this case, the $SO_2$ uptakes on aqueous aerosols are expected to play the dominant roles over dry deposition under specific circumstances such as the haze event (He et al., 2014; Cheng et al., 2016), which should be quantified combining in-situ measurements and

atmospheric modeling.

As shown in Table 2, model schemes often adopt an $\gamma_{eff}$ of $\sim 10^{-4}$ (Liao and Seinfeld, 2005, Wang K et al., 2012), around one order of magnitude higher than the measured values on low RH conditions (Usher et al., 2002; Ullerstam et al., 2003; Adams et al., 2005; Li et al., 2006). For example, in Liao and Seinfeld (2005), $\gamma_{eff}$ is $3.0 \times 10^{-4}$ for RH < 50%, and 0.1 for RH ≥ 50% (see Table 2 with references). At low RH, the uptake coefficient commonly used in model is based on the dry deposition

measurement of $SO_2$ on calcareous soils, cements and $Fe_2O_3$, rather than laboratory measured $\gamma_{eff}$ values that have been recommended by IUPAC. The reason for this divergence is unclear and we are in favor of using the IUPAC recommended $\gamma_{eff}$. The high uptake coefficient in model at high RH is based on two assumptions: fast oxidation of $SO_2$ by $O_3$ in the aqueous phase and high alkalinity in the dust aerosols. Thus this $\gamma_{eff}$ should be applied with caveats that these prerequisites have been fulfilled, especially when extending it for other type of aerosols (Zheng et al. 2015).

**3.4 $N_2O_5$, $HNO_3$, and $H_2O_2$**

$N_2O_5$, $HNO_3$, and $H_2O_2$ demonstrate their high uptake ability on atmospheric aerosols, as shown in Fig. 5. For $N_2O_5$, the similar or higher values of $\gamma_{eff}$ over $\gamma_{eqv}$ demonstrate that the multiphase uptake by all types of aerosols are as important as or even more important than dry deposition. The $N_2O_5$ uptake by aerosols has been widely included in models (Bauer et al., 2004; Liao and Seinfeld, 2005; Wang K et al., 2012). The uptake of $HNO_3$ and $H_2O_2$ by mineral dust and $HNO_3$ by SSA are

important given the overlap between $\gamma_{eff}$ and $\gamma_{eqv}$, thus should also also be detailed characterized in atmospheric models.

For $N_2O_5$, the measured uptake coefficients are $4.8 \times 10^{-3} \sim 0.20$ for mineral dust, $4.0 \times 10^{-5} \sim 6.3 \times 10^{-3}$ for soot, and $6.4 \times 10^{-3} \sim 3.9 \times 10^{-2}$ for SSA, which are comparable to or 1~2 magnitudes higher than the calculated $\gamma_{eqv}$ of $9.3 \times 10^{-4} \sim 2.1 \times 10^{-2}$ under typical conditions (details in Table A.1~A.4). For other kinds of aqueous aerosols, e.g., ammonium sulfate aerosols with high RH, $N_2O_5$ can also be taken up very efficiently with $\gamma_{eff}$ of $10^{-3} \sim 10^{-2}$ (Kane et al., 2001; Schötze and Herrman,

2002; Hallquist et al., 2003; Badger et al., 2006). The importance of $N_2O_5$ and $HNO_3$ uptake by aerosols has been




sufficiently addressed in previous studies (Evans and Jacob, 2005; Liao and Seinfeld, 2005; Stadtler et al., 2018). In current CTMs, $\gamma_{eff}$ of $N_2O_5$ is explicitly calculated as a function of temperature and RH, of which the relation was determined from laboratory experiments (Kane et al., 2001; Bauer et al., 2004; Liao and Seinfeld, 2005).

The extended range of $\gamma_{eqv}$ for $HNO_3$ is $1.5\times10^{-4} \sim 1.5\times10^{-2}$ (urban), $1.5\times10^{-4} \sim 7.7\times10^{-3}$ (agricultural land), $4.2\times10^{-4} \sim$

$3.7\times10^{-1}$ (Amazon) and $7.0\times10^{-4} \sim 7.0\times10^{-2}$ (water), which are within or below the range of $\gamma_{eff}$ for mineral dust and SSA. The higher $\gamma_{eff}$ of $1.0\times10^{-3}$ to 0.21 for mineral dust and of $5.0\times10^{-4}$ to 0.25 for SSA demonstrated a more important role of aerosol uptake than that of the ground surfaces. The uptake of $HNO_3$ on soot and solid organic aerosols appear to be less important. The $HNO_3$ uptake on mineral dust have been implemented in current models with an uptake coefficient of 0.1, or between $1.1 \times 10^{-3}$ and 0.2, consistent with the range of experimentally determined $\gamma_{eff}$ reviewed in this study (Liao and

Seinfeld, 2005; Wang K et al., 2012).

The study on the uptake of $H_2O_2$ by aerosols is rather limited compared to other trace gases aforementioned. The reported $\gamma_{eff}$ on dust and ambient aerosol samples suggest aerosol uptake is more important than that by the ground surface. The measured uptake coefficients of $H_2O_2$ on mineral dust are in the range of $1.0\times10^{-5} \sim 9.4\times10^{-4}$, overlapped with the calculated $\gamma_{eqv}$ of $1.5\times10^{-4} \sim 3.0\times10^{-3}$ under typical conditions. Ambient aerosols collected in urban area show similar $\gamma_{eff}$ of $H_2O_2$ ($8.1\times10^{-5} \sim$

$4.6\times10^{-4}$) to mineral dust (Wu et al., 2015). The aerosol chemistry of $H_2O_2$ in the troposphere is complex and unclear (Liang et al., 2013; Li et al., 2016). In some cases, a net emission of $H_2O_2$ from aerosol surfaces was speculated instead of an uptake or adsorption as a result of $HO_x$ radicals cycling (Liang et al., 2013; Li et al., 2016). Most models only parameterize the $H_2O_2$ uptake by dust particles (Dentener et al., 1996; Wang K et al., 2012). The uptake by other aerosol types hasn't been considered due to limited experimental data. More laboratory kinetic measurements are thus needed. Since ambient aerosol

samples show a $\gamma_{eff}$ similar to that of dust particles (Wu et al., 2015; Pradhan et al., 2010ab; Zhou et al., 2016), we suggest adopting the $\gamma_{eff}$ of dust particles and applying it to all aerosol types before new kinetic data become available.

## 4. Discussion

In this section, we address several important issues based on the comparisons. Large variability found in the measured $\gamma_{eff}$ for $SO_2$ and $NO_2$ are discussed in Sect. 3.5.1. How to apply the measured $\gamma_{eff}$ in atmospheric models to represent the reactivity of

heterogeneous reactions still remains an open question. Regarding this, we discuss the underlying important factors that should be taken into account in Sect. 3.5.2. Outlooks and limitation of this work are provided in Sect. 3.5.3.



## 4.1 Large variability of $\gamma_{eff}$ for $SO_2$ and $NO_2$

Notably, there is a large variability in the reviewed $\gamma_{eff}$ of $SO_2$ uptake by dust particles (as discussed in Sect. 3.2). For $SO_2$ uptake by dust particles, more than three orders of magnitude of differences are found for its uptake by mineral dust ($10^{-8} \sim 10^{-4}$, steady state), which may be attributed to several factors such as the experimental particle substrates, co-existing oxidants ($O_3$, $H_2O_2$, $NO_2$), RH, measurement techniques and surface area used in data processing (Ullerstam et al., 2003; Li et al., 2006; Huang et al., 2015). For example, a $\gamma_{eff}$ of $1.6 \times 10^{-4}$ was derived for $SO_2$ uptake on $Al_2O_3$ powder (Usher et al., 2002). The uptake coefficient was reduced by one order of magnitude to $1.6 \sim 6.6 \times 10^{-5}$ using ambient aerosols of Chinese loess / Saharan dust (Usher et al., 2002; Ullerstam et al., 2003; Adams et al., 2005), indicating that the particle substrate is key in investigating $SO_2$ uptake. Similarly, through cross comparisons between other different investigations shown in Table A.1, we anticipate that the above factors can all contribute to this large discrepancy. As recommended by IUPAC, an uptake coefficient of $4 \times 10^{-5}$ based on airborne measurements is suggested to use in models on low RH conditions. For high RH, we suggest determining $\gamma_{eff}$ with information of aerosol pH because of the high correlation between them as illustrated in Sect. 3.3.

For $NO_2$ uptake on liquid aerosol droplets, three orders of magnitudes of differences are found ($10^{-7} \sim 10^{-4}$), varying significantly with aerosol compositions. On pure water, the uptake is measured at $10^{-7} \sim 10^{-6}$ (Lee and Tang, 1988; Kleffmann et al., 1998; Gutzwiller et al., 2002). On liquid organic aerosols and sea salt aerosols, the uptake can be effectively accelerated to $10^{-6} \sim 10^{-4}$ through multiphase reactions (Abbatt and Waschewsky, 1998; Ammann et al., 2005; Yabushita et al.,2009). For ammonium sulfate aerosols, large discrepancies of $10^{-6} \sim 10^{-4}$ for the initial $\gamma_{eff}$ are found with reasons unexplained yet (Harrison and Collins, 1998; Tan et al., 2016). Based on the reviewed measurements, we suggest using a relatively high uptake coefficient ($\sim 10^{-4}$) for aqueous aerosols containing reactants, and a lower value ($< 10^{-6}$) for other cases.

## 4.2 Initial vs steady state, geometric vs BET

Measurements of effective uptake coefficients revealed the instantly fast uptake at the initial state and gradually declined due to the saturation of surface reaction sites and loss of reactive substances (Hanisch and Crowley, 2003). The uptake at the initial state can be faster of orders of magnitudes higher than that at the steady state for aerosols (see Table 1 and Table A.1~A.4). The time scale reaching surface saturation/equilibrium is dependent on the reaction system. For gas-aqueous particle surface, the timescale to establish equilibrium for the investigated species is less than 1s (Seinfeld and Pandis, pp 554-557, 2006). For dust particles, it can take hours for complete saturation (Judeikis et al., 1978; Goodman et al., 2001). Fine particles with diameters <10 μm have lifetimes of several days in the atmosphere (Prospero, 1999; Lee et al., 2009). Thus using uptake coefficients at steady state maybe more representative in models, unless we can assume that the uptake process is not limited by surface accommodation and reactions (like $HNO_3$, Goodman et al., 2000), typically when the gas



concentration is low enough so the surface passivation is negligible compared to the lifetime of aerosols in the atmosphere (Hanisch and Crowley, 2003). Gas uptake on fresh aerosols may reach or even surpass the level of the ground near emitting sources. Using a uniform uptake coefficient in atmospheric models may not be enough to reflect the deactivation process of the multiphase gas uptake during aerosol aging, considering the large range of $\gamma_{eff}$ varying with time.

In addition, $\gamma_{eff}$ are measured and reported based on the geometric surface or/and the BET surface. More than three orders of magnitudes of differences are derived by whether to consider the pores within the microstructure of solid aerosol surface or not (see Table A.1). In this study, $\gamma_{eff}$ with revised BET surface are generally used as the lower limit, and those using the geometric surface as the upper limit. Whether using BET area as a correction in the calculation of $\gamma_{eff}$ or not remains discrepancy when applied in models (Hanisch and Crowley, 2001ab; Underwood et al., 2001ab). This discrepancy from

measurements may come from the differences in experimental samples (airborne particles vs powder). To solve this issue, more studies on the reactive surface area for ambient aerosols are needed to guide the data processing and model parameterization.

### 4.3 Outlooks and limitations

We can conclude that phase state is a crucial factor influencing the uptake rates. The uptake rates of $O_3$ and $NO_2$ in liquid

organic aerosols are 1~3 orders of magnitudes higher than on solid / semi-solid surfaces. In regions with high RH and sufficient source of organic compounds (e.g., Amazon forest, southern China), the gas uptake is anticipated to have considerable effect on concentrations. The effect is yet to be evaluated combined with further model simulations.

Measurement of uptake by ambient aerosols is crucial to reconcile lab experiments and modeling results, especially for gas with limited investigation conducted (e.g., $H_2O_2$). Currently limited work has been done to address the uptake of $H_2O_2$ by

aerosol particles other than mineral dust (Liao and Seinfeld, 2005; Pradhan et al., 2010 ab; Wang K et al., 2012; Zhou et al., 2016). Because ambient aerosol samples show a $\gamma_{eff}$ comparable to that of dust particles, we recommend similar gamma values of $1.0 \times 10^{-5}$~$9.4 \times 10^{-4}$ for $H_2O_2$ uptake by other types of aerosol, which will lead a larger sink in the atmospheric budget of $H_2O_2$.

Considering the complexity of multiple factors affecting the uptake rates, such as temperature, RH, gas concentration,

aerosol pH, and aerosol state (fresh or aged), establishing a look-up table for $\gamma_{eff}$ based on available factors above should be a feasible way to implement the gas uptake processes in atmospheric models (Mu et al., 2018).

There are limitations for the comparisons conducted in this study. We mainly focused on the uptake fluxes at room temperature (~298K). The gas uptakes at very low temperature (e.g., polar region, stratosphere) are out of scope of this study but should be further explored concerning its potentially large impact. The ambient parameters to calculate the dry

deposition velocities (temperature, radiation) refer to the standard meteorological database for construction in northern China (Zhang, 2004), which may introduce uncertainties for analyses of other areas. The real ambient multiphase processes are



much more complex than the laboratory measurements nevertheless they use airborne aerosols. Ambient on-line measurements of $\gamma_{eff}$ will favor the model parameterization and improve our understanding of the multiphase processes within PBL in the real world (Li et al., 2019). Moreover, more gaseous and aerosol species such as VOCs and bioaerosols should also be investigated (Zhou et al., 1996; Wagner et al., 2002; Fried et al., 2003; Beck et al., 2013; Li et al., 2014; Li et al. 2016; Ouyang et al., 2016; Liu et al. 2017; Meusel et al. 2017).

## 5. Conclusions

In this work, we investigated the relative importance of gas uptake fluxes on ground and aerosols for six reactive trace gases ($O_3$, $NO_2$, $SO_2$, $N_2O_5$, $HNO_3$, $H_2O_2$), various environments, aerosol types and mixing heights. The purpose is to identify aerosol uptake process which is equally or more important than the dry deposition on ground surfaces but has not been adequately addressed in models.

For efficient comparison, we derived a criterion, $\gamma_{eqv}$, to identify which kind of surface is dominant in gas uptake. For investigated gas species, $\gamma_{eqv}$ generally lie in the magnitude of $10^{-4}$, and can be extended to lower values in polluted areas and/or low dry deposition velocities. Especially, $\gamma_{eqv}$ lie in the range of $10^{-6} \sim 10^{-4}$ in polluted urban environments and $10^{-4} \sim 10^{-1}$ under pristine forest conditions. The effective uptake coefficient ($\gamma_{eff}$) derived from experiments are reviewed and compared with $\gamma_{eqv}$. Notably, the gas uptake by aerosols is comparable and should be considered in models when $\gamma_{eff}$ is equal to or higher than $\gamma_{eqv}$. In urban environments, aerosol uptake is important for all combinations of gases and aerosols, favored by the high particle surface densities. On the contrary, the contribution of aerosol uptake is minor compared to dry deposition for gases in the Amazon forest.

The following gas uptake by aerosols can be as important as the dry deposition processes and should be considered in atmospheric models: $N_2O_5$ on all types of aerosols, $HNO_3$ and $H_2O_2$ on mineral dust, $O_3$ on liquid organic aerosols, and $NO_2$, $SO_2$, $HNO_3$ on sea salt aerosols ($\gamma_{eff} \geq \gamma_{eqv}$). The gas uptake on mineral dust for most gases and sea salt aerosols uptake of $SO_2$ and $NO_2$ have already been parameterized in a series of models. The processes of $H_2O_2$ uptake on mineral dust and $O_3$ on liquid aerosols haven't received enough attention unfortunately. For other combinations of gas species and aerosols, the ground tends to be the dominant surface rather than aerosols to take up trace gases within PBL.

(a) It is indicated that the multiphase processes for $O_3$ on liquid organic aerosols are underestimated in current atmospheric models. For regions with high RH and the existence of organic aerosols at liquid state such as Southern China, Amazon forest and South Africa, the multiphase uptakes of $O_3$ by aerosols are expected to contribute comparable sinking fluxes as dry deposition. Compared to the relatively low uptakes on (semi-) solid organic aerosols, we can conclude that phase state is a crucial factor influencing the uptake rates.





(b) Large uncertainties should be addressed for the comparison results of $SO_2$ and $NO_2$. There are more than three orders of magnitude of variances in $\gamma_{eff}$ for $SO_2$ on mineral dust and $NO_2$ on aqueous aerosols. Under low RH circumstances, dry deposition tends to dominate the gas uptake rather than aerosols. However, for cases in high RH, the contributions of aerosols should be cautiously determined with full considerations of the aerosol component, aerosol pH, etc.

5   (c) $H_2O_2$ uptake on a variety of aerosols is needed to be measured and evaluated. It's shown that the $H_2O_2$ uptake on dust is comparable or even more important than that by the ground surface ($\gamma_{eff} \geq \gamma_{eqv}$). Measurements using ambient aerosols suggest that the uptake on aerosols other than mineral dust should be of similar magnitude.

10   ***Data availability.*** All parameters to calculate $V_d$, the aerosol surface area densities ($A$), and the laboratory measurements of $\gamma_{eff}$ are derived from peer-reviewed literature or publicly available database (as illustrated in the main text).

***Author contribution.*** H.S. and Y.C. designed the research. M.L. performed the research with input from H.S., Y.C., and N.M.. U.P. and G.L. discussed the results. M.L., H.S. and Y.C. wrote the manuscript with input from all-co-authors.

***Competing interests.*** The authors declare no conflict of interests.

***Acknowledgements.*** We acknowledge the National Natural Science Foundation of China (91644218), National Key Research and Development Program of China (Grant 2017YFC0210104), and Guangdong Innovative and Entrepreneurial
20   Research Team Program (2016ZT06N263). This work was supported by the Max Planck Society (MPG). Y.C. also acknowledges the Minerva Program of MPG.



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





**Figure 1. Schematic illustration of gas uptake on the ground and on aerosols in the planetary boundary layer as characterized by resistance models. The relative importance of aerosol uptake and dry deposition on the ground is characterized through comparing the aerosol uptake coefficient ($\gamma_{eff}$) with an equivalent uptake coefficient ($\gamma_{eqv}$) corresponding to the deposition velocity ($V_d$).**


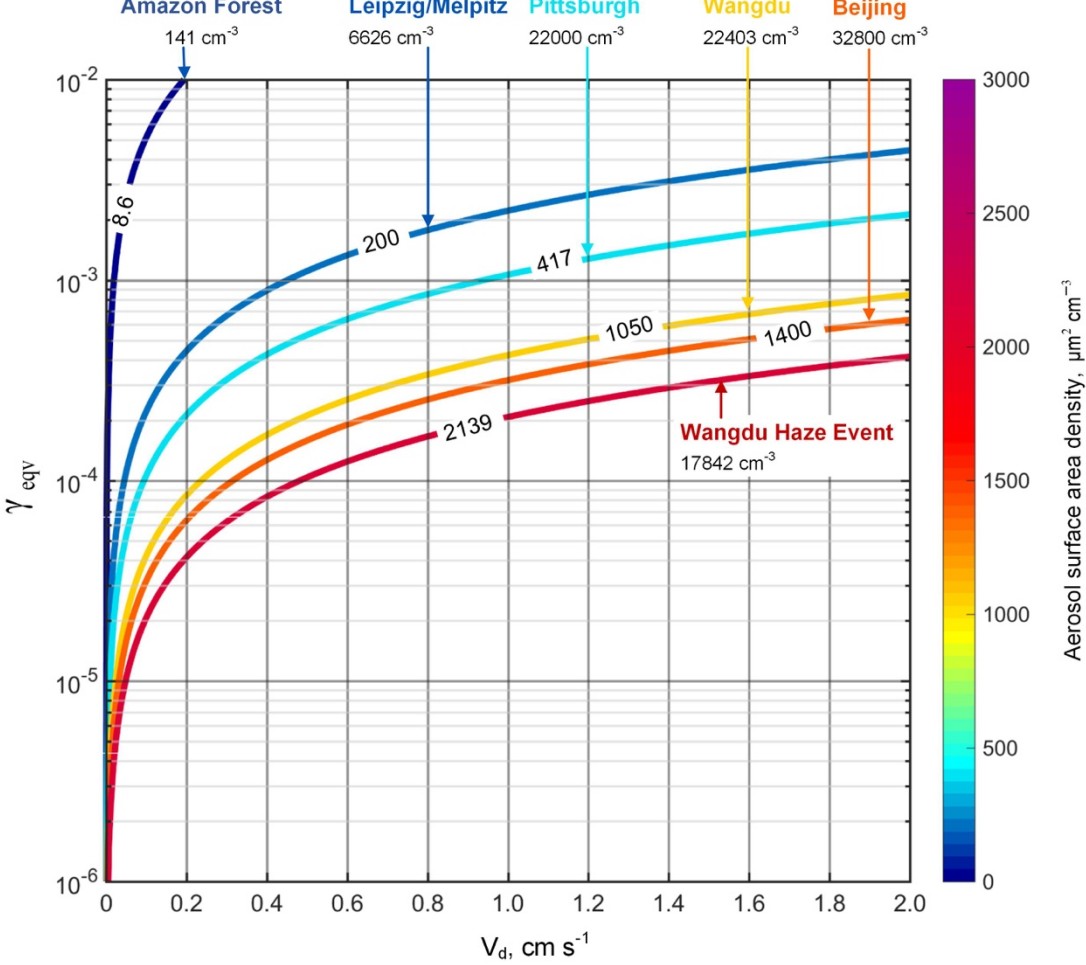

**Figure 2. Relation between $\gamma_{eqv}$ and $V_d$ for a mixing height of 300 m and aerosol surface area densities ($A$) observed at locations and conditions: Amazon Forest (Pöschl et al., 2010), Leipzig/Melpitz (Ma et al., 2014; Ma and Birmili, 2015), Pittsburgh (Stanier et al., 2004), Wangdu with and without haze event (Wu et al., 2017), and Beijing (Wu et al., 2008). For each city/condition, the line is labelled with the corresponding aerosol surface area density. Aerosol particle number concentrations are also provided for orientation.**



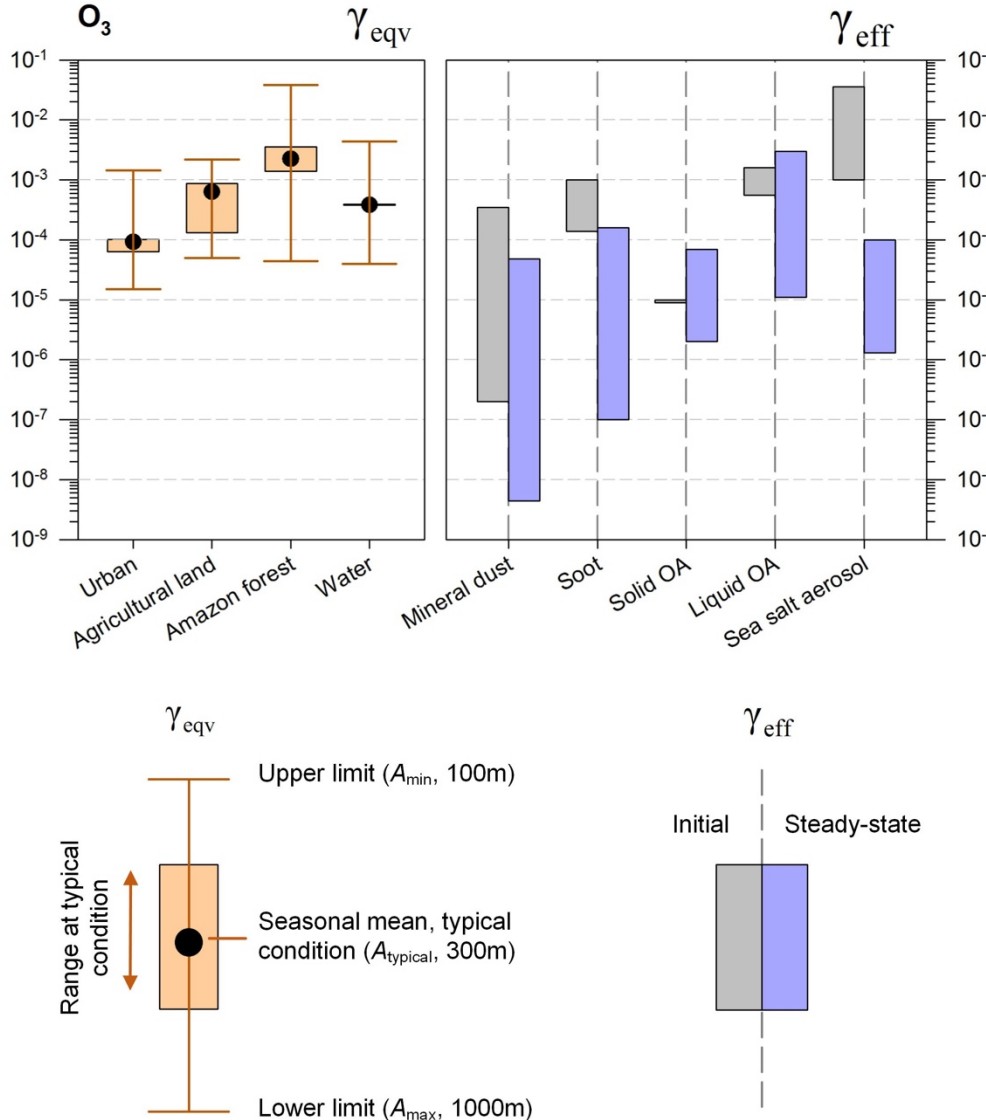

**Figure 3.** Equivalent uptake coefficients ($\gamma_{eqv}$, left) and laboratory measurement values ($\gamma_{eff}$, right) for $O_3$ on different ground types and aerosols.

For $\gamma_{eqv}$, upper whiskers represent maximum values calculated at lowest $A$ and $h$ ($h$ = 100 m), lower whiskers represent minimum
5   values calculated at highest $A$ and $h$ ($h$ = 1 km), and boxes represent typical conditions (typical $A$ as described in Sect. 3.1, $h$ = 300 m). For $\gamma_{eff}$, the grey bar represents the range of initial values, and the purple bar represents the range of steady-state values observed in laboratory experiments.



**Figure 4. Uptake coefficients ($\gamma_{eqv}$, left; $\gamma_{eff}$, right) for NO$_2$ and SO$_2$ on different ground types and aerosols.**





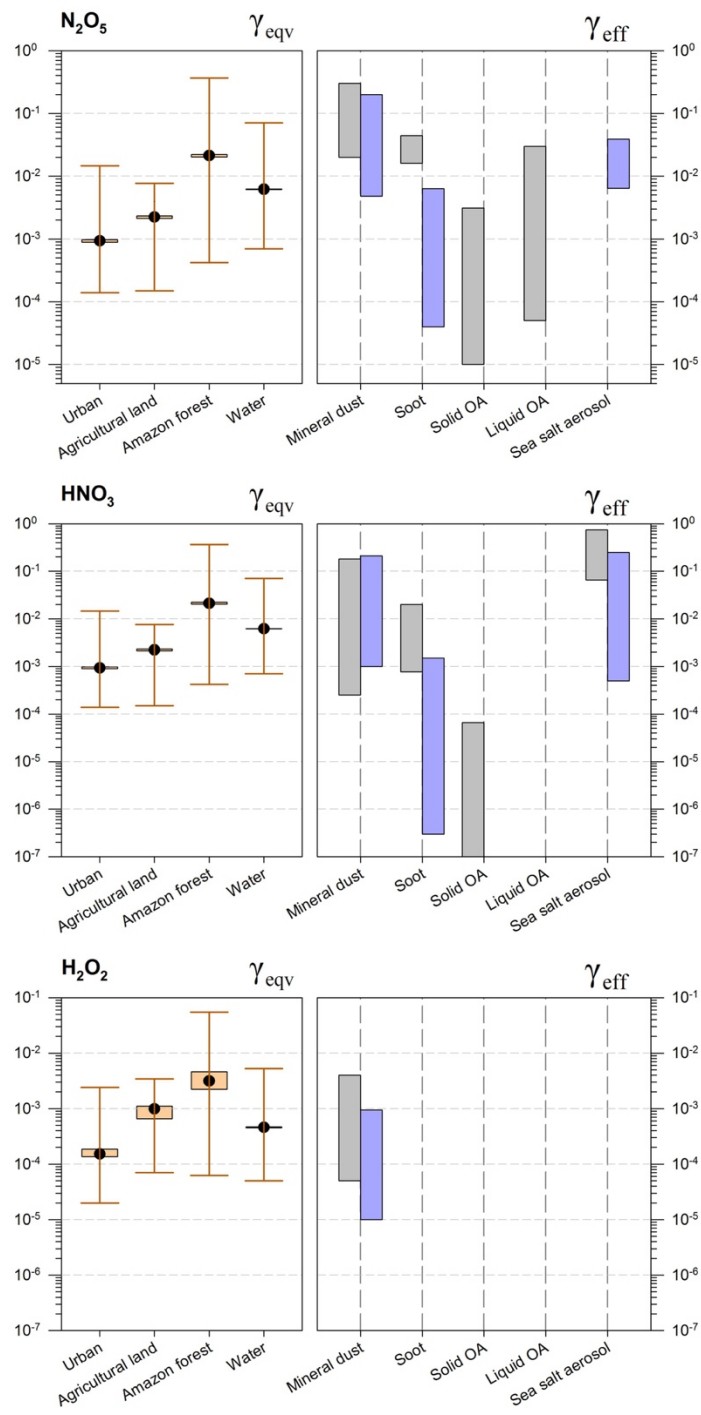

**Figure 5.** Uptake coefficients ($\gamma_{eqv}$, left; $\gamma_{eff}$, right) for $N_2O_5$, $HNO_3$ and $H_2O_2$ on different ground types and aerosols.




**Table 1. Aerosol uptake coefficients (γ$_{eff}$) observed in laboratory experiments [a].**

| Gases | Mineral dust | Soot | Organic aerosol-solid | Organic aerosol-liquid | Sea salt aerosol |
|---|---|---|---|---|---|
| | | | Steady state [b] | | |
| O$_3$ | $4.4\times10^{-9} - 4.8\times10^{-5}$ | $1.0\times10^{-7} - 1.6\times10^{-4}$ | $2.0\times10^{-6} - 6.9\times10^{-5}$ | $1.1\times10^{-5} - 3.0\times10^{-3}$ | $1.3\times10^{-6} - 1.0\times10^{-4}$ |
| NO$_2$ | $1.0\times10^{-9} - 2.3\times10^{-7}$ | $< 5.0\times10^{-8}$ | $<5.0\times10^{-7}$ | $2.2\times10^{-7} - 7.0\times10^{-6}$ | $6.0\times10^{-7} - 3.0\times10^{-4}$ |
| SO$_2$ | $1.5\times10^{-8} - 6.3\times10^{-4}$ | $4.0\times10^{-9} - 2.2\times10^{-6}$ | n/a | n/a | $3.2\times10^{-3} - 1.7\times10^{-2}$ |
| N$_2$O$_5$ | $4.8\times10^{-3} - 2.0\times10^{-1}$ | $4.0\times10^{-5} - 6.3\times10^{-3}$ | n/a | n/a | $6.4\times10^{-3} - 3.9\times10^{-2}$ |
| HNO$_3$ | $1.0\times10^{-3} - 2.1\times10^{-1}$ | $3.0\times10^{-7} - 1.5\times10^{-3}$ | n/a | n/a | $5.0\times10^{-4} - 2.5\times10^{-1}$ |
| H$_2$O$_2$ | $1.0\times10^{-5} - 9.4\times10^{-4}$ | n/a[c] | n/a | n/a | n/a |
| | | | Initial state [b] | | |
| O$_3$ | $2.0\times10^{-7} - 3.5\times10^{-4}$ | $1.4\times10^{-4} - 1.0\times10^{-3}$ | $1.0\times10^{-5}$ | $5.5\times10^{-4} - 1.6\times10^{-3}$ | $1.0\times10^{-3} - 3.6\times10^{-2}$ |
| NO$_2$ | $2.5\times10^{-9} - 2.2\times10^{-5}$ | $1.0\times10^{-6} - 4.0\times10^{-4}$ | $1.0\times10^{-7} - 5.1\times10^{-6}$ | $2.0\times10^{-5}$ | $1.0\times10^{-4}$ |
| SO$_2$ | $1.4\times10^{-7} - 7.7\times10^{-4}$ | $3.0\times10^{-3}$ | n/a | $9.2\times10^{-7} - 1.0\times0^{-5}$ | $6.9\times10^{-3} - 9.0\times10^{-2}$ |
| N$_2$O$_5$ | $2.0\times10^{-2} - 3.0\times10^{-1}$ | $1.6\times10^{-2} - 4.4\times10^{-2}$ | $1.0\times10^{-5} - 3.1\times10^{-3}$ | $5.0\times10^{-5} - 3.0\times10^{-2}$ | n/a |
| HNO$_3$ | $2.5\times10^{-4} - 1.8\times10^{-1}$ | $7.7\times10^{-4} - 2.0\times10^{-2}$ | $\leq 6.6\times10^{-5}$ | n/a | $6.6\times10^{-2} - 7.5\times10^{-1}$ |
| H$_2$O$_2$ | $5.0\times10^{-5} - 4.0\times10^{-3}$ | n/a | n/a | n/a | n/a |

[a] The detailed review table with references of γ$_{eff}$ are in Table A.1~A.4.

[b] The feature (initial or steady state) of the reported uptake coefficients are mainly derived from the literature. If no specific description is found, we assign the measurements on a timescale of ms or s to initial state, and those with longer exposure time (~1h or longer) to steady state.

[c] n/a: not available.




**Table 2. Aerosol uptake coefficients used in atmospheric models[a].**

| Gases | Aerosol type | $\gamma_{eff}$ (Liao and Seinfeld, 2005) | References | $\gamma_{eff}$ (Wang K et al., 2012) | References |
|---|---|---|---|---|---|
| $O_3$ | Mineral dust | $1.0 \times 10^{-5}$ | Michel et al., 2002, 2003 | $5.0 \times 10^{-5} - 1.0 \times 10^{-4}$ | |
| $NO_2$ | Mineral dust | | | $4.4 \times 10^{-5} - 2.0 \times 10^{-4}$ | Underwood et al., 2001 |
| | Wet aerosol | $1.0 \times 10^{-4}$ | Jacob, 2000 | | |
| $SO_2$ | Mineral dust | $3.0 \times 10^{-4}$(RH<50%), 0.1(RH≥50%) | Dentener et al., 1996 | $1.0 \times 10^{-4} - 2.6 \times 10^{-4}$ | Zhang and Carmichael, 1999[b] |
| | Sea salt aerosol | $5.0 \times 10^{-3}$(RH<50%), $5.0 \times 10^{-2}$ (RH≥50%) | Song and Carmichael, 2001[b] | | |
| $N_2O_5$ | Mineral dust | See footnote[c] | Bauer et al., 2004[b] | $1.0 \times 10^{-3} - 0.1$ | Dentener et al., 1996; DeMore et al., 1997 |
| | Organic carbon | $5.2 \times 10^{-4} \times$ RH(RH<50%), 0.03(RH≥50%) | Thornton et al., 2003 | | |
| | Sea salt aerosol | $5 \times 10^{-3}$ (RH<50%), 0.03 (RH≥50%) | Atkinson et al., 2004 | | |
| | Sulfate/nitrate/ ammonium | See footnote [d] | Kane et al., 2001, Hallquist et al., 2003 | | |
| $HNO_3$ | Mineral dust | 0.1 | Hanisch and Crowley, 2001 | $1.1 \times 10^{-3} - 0.2$ | Dentener et al., 1996; DeMore et al., 1997; Underwood et al., 2001 |
| $H_2O_2$ | Mineral dust | | | $1.0 \times 10^{-4} - 2.0 \times 10^{-3}$ | Dentener et al., 1996 |

[a] Here we present two full parameterization schemes: Liao and Seinfeld (2005) and Wang K et al. (2012). The original references of the measurements regarding the uptake coefficients are listed.

5    [b] Model parameterization. The specific references to laboratory measurements are not found.

[c] $\gamma = 4.25 \times 10^{-4} \times RH - 9.75 \times 10^{-3}$

[d] $\gamma = 10^{\beta(T)} \times (C_1 + C_2 \times RH + C_3 \times RH^2 + C_4 \times RH^3)$

$\beta(T) = -4 \times 10^{-2} \times (T - 294), T \geq 282K$

$\beta(T) = 0.48, T < 282K$

10    $C_1 = 2.79 \times 10^{-4}$; $C_2 = 1.30 \times 10^{-4}$; $C_3 = -3.43 \times 10^{-6}$; $C_4 = 7.52 \times 10^{-8}$





**Table A.1. Aerosol uptake coefficients ($\gamma_{eff}$) for reactive gases on mineral dust observed in laboratory experiments (T=298±2K if not specified otherwise).**

| Gases (Unit) | Aerosol type | Initial $\gamma_{eff}$, geometric surface | Initial $\gamma_{eff}$, BET | Steady-state $\gamma_{eff}$, geometric surface | Steady-state $\gamma_{eff}$, BET | References |
|---|---|---|---|---|---|---|
| O$_3$ (×10$^{-5}$) | TiO$_2$/SiO$_2$ | | | 0.3~3 | 0.02~0.32 | Nicolas et al., 2009 |
| | Al$_2$O$_3$, Fe$_2$O$_3$, SiO$_2$ | | 5~18 | | | Michel et al., 2002 |
| | China loess | | 2.7 | | | Michel et al., 2002 |
| | Saharan sand | | 6 | | | Michel et al., 2002 |
| | Al$_2$O$_3$ and others | | 0.27~20 | | 0.6~2.2 | Michel et al., 2003 |
| | Saharan dust | 0.55 ~35 | | 0.22~4.8 | | Hanisch and Crowley, 2003 |
| | Al$_2$O$_3$ | | 0.1~ 1 | | | Sullivan et al., 2004 |
| | Saharan dust | | 0.02~0.6 | | | Chang et al., 2005 |
| | Mineral dust | | | | 0.00044~ 0.01 | Mogili et al., 2006 |
| | Summary | 0.02~35 | | 0.00044~ 4.8 | | |
| NO$_2$ (×10$^{-7}$) | Al$_2$O$_3$ | | | | 0.013~0.26 | Börensen et al., 2000 |
| | Al$_2$O$_3$ and others | | 0.2~220 | | | Underwood et al., 2001b |
| | Saharan dust | | 6.2 | | | Ullerstam et al., 2003 |
| | Illuminated TiO$_2$ | | | 9400, 1200 | | Gustafsson et al., 2006 |
| | CaCO$_3$ | 0.656 | 0.025~ 0.043 | | | Li et al., 2010 |
| | Kaolinite, pyrophylite | | | | 0.07~0.81, 2.3 | Angelini et al., 2007 |
| | TiO$_2$/SiO$_2$, Saharan sand and others | | | | 0.01 | Ndour et al., 2008 |
| | Illuminated TiO$_2$/SiO$_2$ | | | | 1.2~19 | Ndour et al., 2008 |
| | Saharan sand | | | | 0.089, 1 | Ndour et al., 2009 |
| | Arizona test dust | | | | 0.06~0.24 | Dupart et al., 2014 |
| | Kaolin | | 0.31~1.44 | | 0.0256~0.0456 | Liu et al., 2015 |
| | Hematite | | 0.186~1.58 | | 0.0123~0.0150 | Liu et al., 2015 |
| | Summary | 0.025~220 | | 0.01~ 2.3 | | |
| SO$_2$ (×10$^{-5}$) | CaCO$_3$, O$_3$ | 77 | 0.014 | 8.1 | 0.0015 | Li et al., 2006 |
| | Saharan dust, O$_3$ | | | 390 | 0.05 | Ullerstam et al., 2002 |
| | Saharan dust | | 1.6 | | | Ullerstam et al., 2003 |
| | Al$_2$O$_3$, MgO | | 9.5, 26 | | | Goodman et al., 2001 |
| | Al$_2$O$_3$ and others | | 7.0 ~ 51 | | | Usher et al., 2002 |




| | | | | | | |
|---|---|---|---|---|---|---|
| | Chinese loess | | 3 | | | Usher et al., 2002 |
| | Saharan dust, $O_3$ | | 6.6 | | | Adams et al., 2005 |
| | $Al_2O_3$ $Fe_2O_3$, MgO | | 40, 55, 100 | | | Judeikis et al, 1978 |
| | $CaCO_3$ powder | | 0.1 | | | Santschi and Rossi, 2006 |
| | $CaCO_3$, $O_3$ | 43.5~65.6 | 0.026~0.039 | 0.54~22.1 | 0.32~13.2 | Zhang et al., 2018 |
| | Asian mineral dust | | | 10.1~21.4 | | Huang et al., 2015 |
| | Tengger desert dust | | | 22.9~39.0 | | Huang et al., 2015 |
| | Arizona test dust | | | 3.5~9.2 | | Huang et al., 2015 |
| | Asian mineral dust, $H_2O_2$ | | | 39.1~54.5 | | Huang et al., 2015 |
| | Tengger desert dust, $H_2O_2$ | | | 37.2~63.1 | | Huang et al., 2015 |
| | Arizona test dust, $H_2O_2$ | | | 4.6~13.1 | | Huang et al., 2015 |
| | Summary | 0.014~77 | | 0.0015~63.1 | | |
| $N_2O_5$ ($\times 10^{-2}$) | Saharan sand | 30 | | 20 | | Karagulian et al., 2006 |
| | Arizona test dust | 20 | | 11 | | Karagulian et al., 2006 |
| | $CaCO_3$ | 12 | | 2.1 | | Karagulian et al., 2006 |
| | Kaolinite | 16~23 | | 2.2~2.4 | | Karagulian et al., 2006 |
| | Saharan sand | 8 | | 1.3 | | Seisel et al., 2005 |
| | Arizona dust | 0.5~1.3 | | | | Wagner et al., 2008 |
| | Saharan sand | 3.7 | | 3.7 | | Wagner et al., 2008 |
| | Arizona dust | 2.2 | | 2.2 | | Wagner et al., 2008 |
| | $CaCO_3$ | 5.0 | | | | Wagner et al., 2008 |
| | $CaCO_3$ | | | 0.48~0.53 | | Wagner et al., 2009 |
| | $CaCO_3$ | | | 1.13~1.94 | | Wagner et al., 2009 |
| | Arizona dust | | | 0.73~0.98 | | Wagner et al., 2009 |
| | Quarz | | | 0.86~0.45 | | Wagner et al., 2009 |
| | Saharan sand | | | | 2 | Tang et al., 2012 |
| | Arizona dust | | | 0.63 | | Tang et al., 2014 |
| | Illite | | | 9.1, 3.9 | | Tang et al., 2014 |
| | Summary | 2~30 | | 0.48~20 | | |
| $HNO_3$ ($\times 10^{-2}$) | Mineral dust | | | 1.7~5.4 | | Umann et al., 2005 |
| | Arizona dust, $CaCO_3$ and $SiO_2$ | 2~11.3 | | | | Vlasenko et al., 2006 |




| | | | | Reference |
|---|---|---|---|---|
| CaCO$_3$ | | | 0.3~21 | Liu et al., 2008 |
| CaCO$_3$ | 6.0~15 | | | Fenter et al., 1995 |
| CaCO$_3$ | | 0.025 | | Goodman et al., 2000 |
| Saharan dust, Arizona dust, CaCO$_3$ | 11, 6, 10~18 | | | Hanisch and Crowley, 2001a |
| Saharan dust | 13.6 | | | Hanisch and Crowley, 2001b |
| Chinese dust | 17.1 | | | Hanisch and Crowley, 2001b |
| Al$_2$O$_3$ and others | | 0.002~0.0097 | | Underwood et al., 2001a |
| Al$_2$O$_3$ and others | | 0.002~0.61 | | Underwood et al., 2001b |
| Fe$_2$O$_3$ | | 0.0015 | | Frinak et al., 2004 |
| Al$_2$O$_3$, Saharan dust | 13, 11 | 13, 11 | | Seisel et al., 2004 |
| CaCO$_3$ | | 0.2 | | Johnson et al, 2005 |
| CaCO$_3$ powder | | 0.7~30 | 0.07~0.2 | Santschi and Rossi, 2006 |
| Summary | 0.025~18 | | 0.1 ~ 21 | |
| Saharan sand | | | 6.20~9.42 | Pradhan et al., 2010b |
| Gobi sand | | | 3.33~6.03 | Pradhan et al., 2010b |
| TiO$_2$ | | | 15, 5 | Pradhan et al., 2010a |
| Arizona test dust | | 1.47~2.71 | 0.557~ 0.995 | Zhou et al., 2016 [a] |
| Inner Mongolia desert dust | | 2.19~3.56 | 0.25~1.31 | Zhou et al., 2016 [a] |
| Xinjiang dust | | 0.446~0.734 | 0.377~0.431 | Zhou et al., 2016 [a] |
| Arizona test dust | | 3.2 | 0.095~0.185 | El Zein et al., 2014 [b] |
| TiO2, dark | | 2.5~40 | | Romanias et al., 2012 |
| TiO2, UV | | | 35 | Romanias et al., 2012 |
| Al$_2$O$_3$ | | 9.0 | | Romanias et al., 2013 |
| Fe$_2$O$_3$ | | 8.6 | | Romanias et al., 2013 |
| Ambient urban aerosol | | | 0.81 ~ 4.63 | Wu et al., 2015 |
| Summary | 0.5~40 | | 0.1~9.42 | |

(Row group 2 label: H$_2$O$_2$ (×10$^{-4}$))

[a] T=253-313K
[b] T= 268 - 320K





**Table A.2. Aerosol uptake coefficients (γ_eff) for reactive gases on soot observed in laboratory experiments (T=298±2K if not specified otherwise).**

| Gases (Unit) | Aerosol type | Initial $\gamma_{eff}$, geometric surface | Initial $\gamma_{eff}$, BET | Steady-state $\gamma_{eff}$, geometric surface | Steady-state $\gamma_{eff}$, BET | References |
|---|---|---|---|---|---|---|
| $O_3$ (×10⁻⁵) | BC | 100 | | | | Rogaski et al., 1997 |
| | Hydrocarbon soot | | | 16 | 0.5 | Longfellow et al., 2000 |
| | Candle soot | | 13.9 | | 0.628 | Il'in, 1991 |
| | Degussa carbon black | | ~100 | | ~0.001 | Disselkamp et al., 2000 |
| | Spark generated | | 22~330 | | | Fendel et al., 1995 |
| | Spark generated | | 0.12 | | 0.01 | Kamm et al., 1999 |
| | Kerosene, toluene soot | | 18 ~ 38 | | | Lelièvre et al., 2004b |
| | Charcoal | | 22 ~ 413 | | 2.7~11.3 | Stephens et al., 1986 |
| | Spark-generated soot coated with benzo[a]pyrene | | | 0.6~2 | | Pöschl et al., 2001 |
| | Summary | 13.9~100 | | 0.01~16 | | |
| $NO_2$ (×10⁻⁵) | Hydrocarbon soot | | 2.9~5.0 | | <0.001 | Lelièvre et al., 2004a |
| | Spark generated | | 0.15~170 | | 0.0016~0.61 | Kirchner et al., 2000 |
| | Hexane soot | 150~1840 | 2.5 ~ 4.72 | | 0.48~1.17 | Al-Abadleh and Grassian, 2000 |
| | Spark generated | | | | <= 0.004 | Saathoff et al., 2001 |
| | Hexane soot, BC | | | | 0.0015~0.0024 | Prince et al., 2002 |
| | Spark-generated, commercial soot | 3~40 | | | | Kalberer et al., 1996 |
| | Ambient soot | 1100 | | 33 | | Ammann et al. 1998 |
| | Commercial soot | | 0.1 | | <0.001 | Kleffmann et al., 1999 |
| | Spark-generated | | 0.5~1.0 | | | Arens et al., 2001 |
| | Summary | 0.1~40 | | <0.005 | | |
| $SO_2$ (×10⁻⁶) | BC | 3000 | | | | Rogaski et al., 1997 |
| | Fresh BC, aged BC | | | | 0.00398, 0.32 | Xu et al., 2015 |
| | BC, $O_3$ | | | | 2.17 | Xu et al., 2015 |
| | Summary | 3000 | | 0.004 ~ 2.17 | | |
| $N_2O_5$ (×10⁻³) | Decane soot | 44 | | 5 | | Karagulian and Rossi, 2007 |
| | Spark generated | | | | 0.04~0.2 | Saathoff et al., 2001 |



| | | | | | | |
|---|---|---|---|---|---|---|
| | Hydrocarbon soot | 16 | | 6.3 | | Longfellow et al., 2000 |
| | Summary | 16~44 | | 0.04~6.3 | | |
| HNO$_3$($\times 10^{-4}$) | Decane soot | 200 | | 4.6~5.2 | | Salgado-Muñoz and Rossi, 2002 |
| | Spark generated | | | | 0.003 | Saathoff et al., 2001 |
| | Hydrocarbon soot | | | 15 | 0.5 | Longfellow et al., 2000 |
| | Spark generated | | 0.052~7.7 | | 0.00098~0.019 | Kirchner et al., 2000 |
| | Summary | 7.7~200 | | 0.003~15 | | |



**Table A.3. Aerosol uptake coefficients ($\gamma_{eff}$) for reactive gases on organic aerosols observed in laboratory experiments (T=298±2K if not specified otherwise).**

| Gases (Unit) | Aerosol type | Initial $\gamma_{eff}$, geometric surface | Initial $\gamma_{eff}$, BET | Steady-state $\gamma_{eff}$, geometric surface | Steady-state $\gamma_{eff}$, BET | References |
|---|---|---|---|---|---|---|
| $O_3$ ($\times 10^{-5}$) | Semi-solid protein aerosol | 1.0 | | < 1.0 | | Shiraiwa et al., 2011 |
| | Shikimic acid film | | | 0.2~1.0 | | Berkemeier et al., 2016 |
| | Solid 1-hexadecene | | | 0.64~2.5 | | Moise and Rudich, 2000[a] |
| | Monolayer organic film | | | 17~27 | | Moise and Rudich, 2000[b] |
| | Solid-liquid oleic acid | | | 2~17 | | Knopf et al., 2005 |
| | Solid-liquid oleic acid (meet-cooking aerosols) | | | 1.6~6.9 | | Knopf et al., 2005 |
| | Liquid oleic acid particle | 150 | | 5 | | Mendez et al., 2014 |
| | Liquid organic compounds | | | 1.0~100 | | de Gouw et al., 1998 |
| | Aqueous α-pinene aerosol | | | 300~750 | | King et al., 2008[c] |
| | Aqueous fumarate aerosol | | | 1.1 | | King et al., 2008[c] |
| | Aqueous benzoate aerosol | | | 1.5 | | King et al., 2008[c] |
| | Liquid oleic acid aerosol | 160 | | | | Morris et al., 2002 |
| | Oleic acid aerosol | 55~90 | | 20~100 | | Sage et al., 2009 |
| | Liquid 1-tridecene | | | 52~55 | | Moise and Rudich, 2000[d] |
| | Liquid 1-hexadecene | | | 32~38 | | Moise and Rudich, 2000[e] |
| | Liquid 1-hexadecane | | | 2.0 | | Moise and Rudich, 2000 |
| | Liquid oleic acid | | | 88 | | Hearn et al., 2005 |
| | Liquid oleic acid | | | 40~72 | | Knopf et al., 2005 |
| | Liquid oleic acid | | | 730 | | Smith et al., 2002 |
| | Summary | ~1.0 for solid, 55~160 for liquid | | 0.2~6.9 for solid, 1.1~300 for liquid | | |
| $NO_2$ ($\times 10^{-6}$) | Soild benzophenone, catechol, anthracene, anthrarobin | 0.07~1.26 (dark), 0.65~2.40 (light) | | | | George et al., 2005[f] |
| | Soild benzophenone, catechol, anthracene, anthrarobin | 0.24 ~ 3.6 (dark), 1.3~5.1 (light) | | | | George et al., 2005[f] |
| | Solid 1,2,10-trihydroxyanthracene | 0.7 ~ 2 | | | | Arens et al., 2002 |
| | Solid 1,2,10-trihydroxyanthracene | | | < 0.5 | | Arens et al., 2002 |
| | Nitroguaiacol, mixture of organics | 52, 22 | | | | Knopf et al., 2011 |
| | Solid levoglucosan, abietic acid | < 1.0 | | | | Knopf et al., 2011 |
| | Solid pyrene, dark, near-UV | | | < 0.1, 3.5 | | Brigante et al., 2008 |
| | Solid pyrene | <= 1.0 | | | | Gross et al., 2008[g] |





| | | | | | |
|---|---|---|---|---|---|
| | SOA, pinene/O$_3$ | | | < 0.5 | | Bröske et al., 2003 |
| | SOA, limonene/O$_3$, catechol/O$_3$, limonene/OH, toluene/OH | | | <1.5 | | Bröske et al., 2003 |
| | Humic acid, light | 20 | | | | Stemmler et al., 2006 |
| | Humic acid, dark | | | < 0.1 | | Stemmler et al., 2007 |
| | Humic acid, illuminated | | | 2.6, 3.7 | | Stemmler et al., 2007 |
| | Catechol (surface-absorbed) + NaCl/NaBr/NaF | | | | 3 ~ 7 | Woodill and Hinrichs, 2010 |
| | Gentisic acid, tannic acid, UV/Vis light | | | | 0.22 ~0.88 | Sosedova et al., 2011 |
| | Solution of guaiacol, syringol, catechol | | | <0.1 for pH <7 , 10 for pH > 10 | | Ammann et al., 2005 |
| | Summary | 0.1~5.1 for solid, 20 for liquid | | <0.5 for solid, 0.22~7 for liquid | | |
| SO$_2$(× 10$^{-6}$) | Liquid oleic acid | 0.92 ~ 6.44 | | | | Shang et al., 2016[c] |
| | Liquid SOA by limonene and O$_3$ | 10 ~50 | | | | Ye et al., 2018 |
| | Summary | 0.92~10 for liquid | | Not available | | |
| N$_2$O$_5$ (×10$^{-3}$) | Solid malonic acid | <1.0 | | | | Thornton et al., 2003 |
| | Solid azelaic acid | 0.5 | | | | Thornton et al., 2003 |
| | Solid oxalic acid | <0.01 | | | | Griffiths et al., 2009 |
| | Solid oxalic acid | 3.1 | | | | Griffiths et al., 2009 |
| | Solid succinic acid | <0.6, <0.3 | | | | Griffiths et al., 2009 |
| | Aqueous malonic acid | 2.0, 30 | | | | Thornton et al., 2003 |
| | Humic acid | 0.1, 0.3, 1.0 | | | | Badger et al., 2006 |
| | Malonic acid | 8 ~ 16 | | | | Griffiths et al., 2009 |
| | Succinic acid | 5.2 ~ 9 | | | | Griffiths et al., 2009 |
| | Glutaric acid | 0.6 ~ 8 | | | | Griffiths et al., 2009 |
| | Summary | 0.01~3.1 for solid, 0.05 ~ 30 for liquid | | Not available | | |
| HNO$_3$ (×10$^{-5}$) | Solid pyrene | ≤ 6.6 | | | | Gross et al., 2008[g] |
| | Summary | ≤ 6.6 for solid | | Not available | | |

[a] T=272K
[b] T=219-298K
[c] T=293K
5  [d] T=272-298K
[e] T=283-298K
[f] T=278-308K
[g] T=293-297K



**Table A.4. Aerosol uptake coefficients ($\gamma_{eff}$) for reactive gases on sea salt aerosols observed in laboratory experiments (T=298±2K if not specified otherwise).**

| Gases (Unit) | Aerosol type | Initial $\gamma_{eff}$, geometric surface | Initial $\gamma_{eff}$, BET | Steady-state $\gamma_{eff}$, geometric surface | Steady-state $\gamma_{eff}$, BET | References |
|---|---|---|---|---|---|---|
| $O_3$ ($\times 10^{-3}$) | Synthetic sea salt | 1.0~10 | | | | Mochida et al., 2000 |
| | Natural sea salt | 0.97 | | | | Mochida et al., 2000 |
| | NaCl | | | | 0.0013 | Il'in et al., 1991[a] |
| | Deliquesced NaCl | | | <0.1 | | Abbatt and Waschewsky, 1998 |
| | NaCl/$Fe_2O_3$ | | 1.3, 33~36 | | | Sadanaga et al., 2001 |
| | Summary | 1.0~ 36 | | 0.0013~0.1 | | |
| $NO_2$ ($\times 10^{-4}$) | Deliquesced NaCl | <1.0 | | | | Abbatt and Waschewsky, 1998 |
| | Deliquesced NaCl | | | 2.8~3.7 | | Harrison and Collins, 1998[b] |
| | Aqueous NaCl | 1.0 | | | | Yabushita et al.,2009 |
| | Chinese seasalt | | | | 0.00551 | Ye et al., 2010 |
| | Chinese seasalt | | | | 0.0126 | Ye et al., 2010 |
| | NaCl | | | | 0.6 | Vogt et al., 1994 |
| | Summary | 1.0 | | 0.006~3.0 | | |
| $SO_2$ ($\times 10^{-3}$) | Synthetic sea salt | 6.0~90 | | 3.2~17 | | Gebel et al., 2000 |
| | Summary | 6.0~90 | | 3.2~17 | | |
| $N_2O_5$ ($\times 10^{-2}$) | NaCl | | | | 3.2 | Behnke et al., 1997 |
| | NaCl | | | | 0.64 | Stewart et al., 2004 |
| | NaCl | | | | 0.9 | Stewart et al., 2004 |
| | NaCl | | | | 1.04 | Stewart et al., 2004 |
| | NaCl | | | | 0.078 | Stewart et al., 2004 |
| | Sea salt | | | | 1.6 | Stewart et al., 2004 |
| | Sea salt | | | | 2.8 | Stewart et al., 2004 |
| | Sea salt | | | | 1.3 | Stewart et al., 2004 |
| | Sea salt | | | | 3.1 | Stewart et al., 2004 |
| | Synthetic sea salt | | | | 2.2 | Thornton and Abbatt, 2005 |
| | Synthetic sea salt | | | | 3.0 | Thornton and Abbatt, 2005 |
| | Synthetic sea salt | | | | 2.4 | Thornton and Abbatt, 2005 |
| | NaCl | | | | 1.8 | Schweitzer et al., 1998[c] |
| | NaCl | | | | 1.4~3.9 | George et al., 1994 |
| | Synthetic sea salt | | | 0.29 | | Hoffman et al., 2003 |
| | Summary | Not available | | 0.64~3.9 | | |
| $HNO_3$ ($\times 10^{-2}$) | Deliquesced sea salt | 50 | | | | Guimbaud et al., 2002 |
| | Synthetic sea salt | 6.6~75 | | 3.0~25 | | De Haan and Finlayson-Pitts, 1997 |



| | | | | |
|---|---|---|---|---|
| deliquesced NaCl, 100nm size | 0.49 | | | Tolocka et al., 2004 |
| Deliquesced NaCl | 15 | | | Saul et al., 2006 |
| Deliquesced NaCl | 21~11 | | | Liu et al., 2007 |
| NaCl/MgCl2 | 25~12 | | | Liu et al., 2007 |
| Sea salt | 27~12 | | | Liu et al., 2007 |
| Deliquesced NaCl | 20 | | | Abbatt and Waschewsky, 1998 |
| Seliquesced NaCl | 50 | | | Stemmler et al., 2008 |
| Synthetic sea salt | | | 0.04~0.065 | Hoffman et al., 2003 |
| Summary | 6.6~75 | | 0.05-25 | |

[a] T=235-299K
[b] T = 279K
[c] T=262-278K

