# Peer review of "Relative importance of gas uptake on aerosol and ground surfaces characterized by equivalent uptake coefficients"

_Atmospheric Chemistry and Physics, 2019_

## Referee Comment (RC1) · Mingjin Tang (Referee) · 21 Apr 2019

I know some of the authors very well, although such acquaintance does not preclude me being objective as a referee; in addition, I have asked the authors to consider if some of my papers should be cited. Therefore, I chose to disclose my name as a referee to make the review process more transparent.

In this work, Li et al. proposed "equivalent uptake coefficient", used this term to compare the relative importance of gas uptake onto aerosol surface versus group surface, and concluded that some uptake processes onto aerosol particles can be very important. The methodology is novel, and the results can be interesting for the atmospheric chemistry community. The manuscript can be accepted after the following comments are addressed.

P41, Table 1: There are some experimental studies (by Joel A Thornton, Jon Abbatt, Tim Bertram, and likely other) which explored the effect of organics on N2O5 uptake. In addition, there may be more studies on H2O2 uptake. Please check the IUPAC evaluation online as well as relevant literature.

P42, Table 2: I am not sure why the work by Wang et al. (2012) is used a representative example here. In fact the uptake coefficients used by Wang et al. are far from being updated, and they mainly used uptake coefficients adopted by two modeling studies almost 20 years ago (Dentener et al., 1996; Zhang and Carmichael, 1999). For mineral dust in specific, the uptake coefficients used by Zhu et al. (2010) were updated values recommended by IUPAC. In addition, some of the studies which are cited as the sources of uptake coefficients measured by laboratory work are in fact pure modeling work, such as Bauer et al. (2004), Dentener et al. (1996), and so on. The author may consider updating this table.

P3, L23-25: HO2 uptake can be very important for tropospheric chemistry (George et al., 2013; Mao et al., 2013; Taketani et al., 2008; Thornton et al., 2008). Is there a reason why HO2 has not been discussed in this paper?

P2, L29-30: Very recently I reviewed heterogeneous reactions of mineral dust (Tang et al., 2017). Should this paper be cited here?

P3, L12-15: Another convenient way to assess the relative importance of aerosol uptake and dry deposition is to calculate their lifetimes with respect to individual processes, as discussed by Tang et al. (2017)

P4, L24-26: This sentence is not easy to follow. I assume that the authors wanted to state

that for smaller particles, gas phase diffusion would not be a limiting step and thus can be neglected. Please consider rephrasing it, and refer to Tang et al. (2014) for a comprehensive discussion on the role of gas phase diffusion.

P12-13: In a paper published in 2017 (Tang et al., 2017), I provided a comprehensive and in-depth discussion on the two factors the authors mentioned in Section 4.2, and would like to refer the authors to take a look at that paper.

P14, L19-21 (as well as related content in the abstract): It is proposed that the following four groups of gas uptake onto aerosols can be important: (1) $N_2O_5$ on all types of aerosols, (2) $HNO_3$ and $H_2O_2$ on mineral dust, (3) $O_3$ on liquid organic aerosols; and (4) $NO_2$, $SO_2$, $HNO_3$ on sea salt aerosols. The four groups have some overlaps and not easy to follow. I would suggest re-organizing them according to either types of gases or types of aerosols.

**References:**

George et al.: Measurements of uptake coefficients for heterogeneous loss of HO2 onto submicron inorganic salt aerosols, Phys. Chem. Chem. Phys., 15, 12829-12845, 2013.

Mao et al.: Radical loss in the atmosphere from Cu-Fe redox coupling in aerosols, Atmos. Chem. Phys., 12, 509-519, 2013.

Tang et al.: Compilation and evaluation of gas phase diffusion coefficients of reactive trace gases in the atmosphere: volume 1. Inorganic compounds, Atmos. Chem. Phys., 14, 9233-9247, 2014.

Tang et al.: Heterogeneous reactions of mineral dust aerosol: implications for tropospheric oxidation capacity, Atmos. Chem. Phys., 17, 11727-11777, 2017.

Taketani et al.: Kinetics of heterogeneous reactions of HO2 radical at ambient concentration levels with (NH4)2SO4 and NaCl aerosol particles, J. Phys. Chem. A, 112, 2370-2377, 2008.

Thornton et al.: Assessing known pathways for HO2 loss in aqueous atmospheric aerosols: Regional and global impacts on tropospheric oxidants, J. Geophys. Res.-Atmos, 113, D05303, doi: 05310.01029/02007JD009236, 2008.

Zhu, S., Butler, T., Sander, R., Ma, J., and Lawrence, M. G.: Impact of Dust on Tropospheric Chemistry over Polluted Regions: a Case Study of the Beijing Megacity, Atmos. Chem. Phys., 10, 3855-3873, 2010.

---

## Referee Comment (RC2) · Anonymous Referee #2 · 1 May 2019

The manuscript "Relative importance of gas uptake on aerosol and ground surfaces characterized by equivalent uptake coefficients" presented a theoretical approach to characterize the relative importance of uptake of trace gases on aerosols versus on ground. The authors proposed a new parameter "equivalent uptake coefficient" ($\gamma_{eqv}$) at which the flux of gas uptake on aerosols is equal to that on ground and derived $\gamma_{eqv}$ under various environment (vertical velocity and particle surface concentration). By comparing $\gamma_{eqv}$ with the effective uptake coefficient of gases on aerosols ($\gamma_{eff}$) reviewed from literature, the authors assessed the relative importance of gas uptake on aerosols to dry deposition. It was found that under urban environment, gas uptake on all types of aerosols (mineral dust, sea salt, organic aerosol, and soot) is important, while in pristine Amazonia forest the contribution of uptake on aerosols to gas loss is minor. $N_2O_5$ uptake on all types aerosol, $HNO_3$ and $H_2O_2$ on mineral aerosols, $O_3$ on liquid organic aerosol, $NO_2$, $SO_2$ and $HNO_3$ on sea salt aerosol are as important as dry deposition. The author also pointed out that $H_2O_2$ uptake on various aerosols need further laboratory studies and to be evaluated.

The approach presented is a novel and convenient way to compare the relative importance of uptake of gases on aerosols with dry deposit. This manuscript is well written and easy to follow. And the discussion is well balanced. I have only a few minor comments, mainly to clarify some discussion. I recommend the direct publication of this manuscript on ACP after these minor comments are fixed.

1.  Pg. 4 line 18, a typical value of $\omega$ of 300 m$^{-1}$ is used. I understand this can simplify the equation and $\gamma_{eqv}$, since different gases have slightly different mean velocity, especially in order to get a clear picture as shown in Fig. 2. Are the $\gamma_{eqv}$ values in Fig. 3-5 also calculated in this way? It might be helpful to briefly mention the influence of this simplification in the discussion part "Sect. 4.3".

2.  Pg. 10 line 11, I am curious why the authors mainly discussed the model schemes in the studies Liao and Seinfeld (2005) and Wang K et al. (2012) among other model studies including heterogeneous reactions.

3.  Pg. 11 line 24,"…Sect. 3.5.1…", I guess that the authors meant "4.1.1". Also check line 26.

4.  Pg. 13 line 27-Pg. 14 line 5, it might be helpful to also mention that the variability of aerosol surface concentration under each environment could also contribute to the variability of $\gamma_{eqv}$.

5.  Pg. 14 line 25, it seems that one leading sentence is missing before "(a)…". Please double check.

6.  Pg. 14 line 20, "…$HNO_3$ and $H_2O_2$ on mineral…", according to Fig. 2 should $SO_2$ be also listed here?

7.  Pg. 38 line 6, "…the purple bar…" should be "blue bar".

---

## Author Comment (AC1) · 9 Jul 2019

**Response to referee #1**

*In this work, Li et al. proposed "equivalent uptake coefficient", used this term to compare the relative importance of gas uptake onto aerosol surface versus group surface, and concluded that some uptake processes onto aerosol particles can be very important. The methodology is novel, and the results can be interesting for the atmospheric chemistry community. The manuscript can be accepted after the following comments are addressed.*

**Response:** We thank the positive and constructive comments given by the referee #1, which are very helpful to improve the manuscript. Our response to each specific comment is presented below.

Detailed Comments and Responses:

*1. P41, Table 1: There are some experimental studies (by Joel A Thornton, Jon Abbatt, Tim Bertram, and likely other) which explored the effect of organics on $N_2O_5$ uptake. In addition, there may be more studies on $H_2O_2$ uptake. Please check the IUPAC evaluation online as well as relevant literature.*

**Response:** We thank the referee's comments. In the last version, we have already included the following experimental studies suggested by the referee in Table A.3 regarding $N_2O_5$ uptake on organics, i.e., Thornton et al. (2003), Griffiths et al. (2009), and Badger et al. (2006). In the revised manuscript, we have tried to complete the list by including Folkers et al. (2003), Gross et al. (2009) and Anttila et al. (2006) as in the new Table A.3.

For $H_2O_2$, we checked the IUPAC evaluation online data and related literature. We now add one more measurement of $H_2O_2$ uptake on mineral dust (Zhao et al., 2011) in the revised manuscript. We are still unable to find more laboratory measurements of $H_2O_2$ uptake on aerosols other than mineral dust, thus more measurements are needed in the future.

*2. P42, Table 2: I am not sure why the work by Wang et al. (2012) is used a representative example here. In fact, the uptake coefficients used by Wang et al. are far from being updated, and they mainly used uptake coefficients adopted by two modeling studies almost 20 years ago (Dentener et al., 1996; Zhang and Carmichael, 1999). For mineral dust in specific, the uptake coefficients used by Zhu et al. 2010 were updated values recommended by IUPAC. In addition, some of the studies which are cited as the sources of uptake coefficients measured by laboratory work are in fact pure modeling work, such as Bauer et al. (2004), Dentener et al. (1996), and so on. The author may consider updating this table.*

**Response:** We thank the referee's comments. The scheme of Wang K et al. (2012) is taken as an example considering the large impact/applications of this scheme within the community (e.g., Wang et al., 2014; Li et al., 2015; Zheng et al., 2015). We update the table in the revised manuscript by including the scheme of Zhu et al. (2010) which uses updated values recommended by IUPAC. It should be addressed that we only provide examples of model

schemes that have considered the heterogeneous reactions to give an overall implication for modelers, rather than to give a complete overview of the parameterizations of uptake coefficients covering all modeling studies, which is out of the scope of this paper. To avoid misunderstanding, we revise the caption to "Examples of aerosol uptake coefficients used in atmospheric models", add more illustrations in the footnote and main text, and move Table 2 to the supplement.

Uptake coefficients with sources listed as Dentener et al. (1996), Bauer et al. (2004), Song and Carmichael (2001) in the table are from model parameterization, and the specific laboratory measurements are not found in the literature. We have already noted this issue in the last version (see footnote [b]).

*3. P3, L23-25: HO$_2$ uptake can be very important for tropospheric chemistry (George et al., 2013; Mao et al., 2013; Taketani et al., 2008; Thornton et al., 2008). Is there a reason why HO$_2$ has not been discussed in this paper?*

**Response:** We agree that HO$_2$ uptake on aerosol is important for atmospheric chemistry. The motivation of our work is to compare the fluxes of dry deposition and aerosol uptake, which is difficult for radicals like HO$_2$ because the required parameters to calculate dry deposition are not available from current literature. The reason could be that the other pathways are too fast compared to the deposition of HO$_2$.

*4. P2, L29-30: Very recently I reviewed heterogeneous reactions of mineral dust (Tang et al., 2017). Should this paper be cited here?*

**Response:** We have included this review paper in the revised manuscript along with other original research articles.

*5. P3, L12-15: Another convenient way to assess the relative importance of aerosol uptake and dry deposition is to calculate their lifetimes with respect to individual processes, as discussed by Tang et al. (2017).*

**Response:** We agree with the referee that comparison of lifetimes between aerosol uptake and dry deposition is another feasible method, as discussed in Tang et al. (2017). The method proposed in our study share the same basic formulations with Tang et al. (2017), and velocities can be easily converted to lifetime.

*6. P4, L24-26: This sentence is not easy to follow. I assume that the authors wanted to state that for smaller particles, gas phase diffusion would not be a limiting step and thus can be neglected. Please consider rephrasing it, and refer to Tang et al. (2014) for a comprehensive discussion on the role of gas phase diffusion.*

Response: We thank the referee's comments. We have referred to the original formation of Jacob 2000 as follows:

"For atmospheric aerosols with a diameter of ~0.2 μm or smaller, the related gaseous uptake tends to be limited by the free molecular collision rate (uptake rate $\rightarrow \omega\alpha A[X_g]/4$) (Jacob, 2000). Thus in the following analyses, we mainly focus on the discussion of $\gamma_{eff}$, and neglect the diffusion resistance in the gas phase."

*7. P12-13: In a paper published in 2017 (Tang et al., 2017), I provided a comprehensive and in depth discussion on the two factors the authors mentioned in Section 4.2, and would like to refer the authors to take a look at that paper.*

**Response:** Thanks for the referee's comments. We refer to this work in our discussion as follows:

"More than three orders of magnitudes of differences are derived by whether to consider the pores within the microstructure of solid aerosol surface or not (see Table A.1). Using the same method to calculate the available surface area may reconcile these differences (Tang et al., 2017)."

*8. P14, L19-21 as well as related content in the abstract It is proposed that the following four groups of gas uptake onto aerosols can be important: 1) $N_2O_5$ on all types of aerosols, 2) $HNO_3$ and $H_2O_2$ on mineral dust, 3) $O_3$ on liquid organic aerosols; and 4) $NO_2$, $SO_2$, $HNO_3$ on sea salt aerosols. The four groups have some overlaps and not easy to follow. I would suggest re organizing them according to either types of gases or types of aerosol s*

**Response:** This is re-organized to present the most intensive summary of our conclusion.

We will add the following table to better illustrate it. According to Table 2, there is no overlapped information and four bullets are already the minimum number of vectors from this matrix.

**Table 2. Gas uptake processes that are potentially important compared to dry deposition across various environments (marked with √).**

| Gases | Mineral dust | Soot | Organic aerosol-solid | Organic aerosol-liquid | Sea salt aerosol |
|-------|------------|------|---------------------|----------------------|-----------------|
| $O_3$ | | | | √ | |
| $NO_2$ | | | | | √ |
| $SO_2$ | √ | | | | √ |
| $N_2O_5$ | √ | √ | √ | √ | √ |
| $HNO_3$ | √ | | | | √ |
| $H_2O_2$ | √ | | | | |

**References**

Anttila, T., Kiendler-Scharr, A., Tillmann, R., and Mentel, T. F.: On the Reactive Uptake of Gaseous Compounds by Organic-Coated Aqueous Aerosols: Theoretical Analysis and Application to the Heterogeneous Hydrolysis of N2O5, The Journal of Physical Chemistry A, 110, 10435-10443, 10.1021/jp062403c, 2006.

Badger, C. L., Griffiths, P. T., George, I., Abbatt, J. P. D., and Cox, R. A.: Reactive Uptake of N2O5 by Aerosol Particles Containing Mixtures of Humic Acid and Ammonium Sulfate, The Journal of Physical Chemistry A, 110, 6986-6994, 10.1021/jp0562678, 2006.

Folkers, M., Mentel, T. F., and Wahner, A.: Influence of an organic coating on the reactivity of aqueous aerosols probed by the heterogeneous hydrolysis of N2O5, Geophys. Res. Lett., 30, 10.1029/2003GL017168, 2003.

Griffiths, P. T., Badger, C. L., Cox, R. A., Folkers, M., Henk, H. H., and Mentel, T. F.: Reactive Uptake of N2O5 by Aerosols Containing Dicarboxylic Acids. Effect of Particle Phase, Composition, and Nitrate Content, The Journal of Physical Chemistry A, 113, 5082-5090, 10.1021/jp8096814, 2009.

Gross, S., Iannone, R., Xiao, S., and Bertram, A. K.: Reactive uptake studies of NO3 and N2O5 on alkenoic acid, alkanoate, and polyalcohol substrates to probe nighttime aerosol chemistry, Physical Chemistry Chemical Physics, 11, 7792-7803, 10.1039/B904741G, 2009.

Li, X., Zhang, Q., Zhang, Y., Zheng, B., Wang, K., Chen, Y., Wallington, T. J., Han, W., Shen, W., Zhang, X., and He, K.: Source contributions of urban PM2.5 in the Beijing–Tianjin–Hebei region: Changes between 2006 and 2013 and relative impacts of emissions and meteorology, Atmospheric Environment, 123, 229-239, https://doi.org/10.1016/j.atmosenv.2015.10.048, 2015.

Thornton, J. A., Braban, C. F., and Abbatt, J. P. D.: N2O5 hydrolysis on sub-micron organic aerosols: the effect of relative humidity, particle phase, and particle size, Physical Chemistry Chemical Physics, 5, 4593-4603, 10.1039/B307498F, 2003.

Wang, K., Zhang, Y., Nenes, A., and Fountoukis, C.: Implementation of dust emission and chemistry into the Community Multiscale Air Quality modeling system and initial application to an Asian dust storm episode, Atmos. Chem. Phys., 12, 10209-10237, 10.5194/acp-12-10209-2012, 2012.

Wang, Y., Zhang, Q., Jiang, J., Zhou, W., Wang, B., He, K., Duan, F., Zhang, Q., Philip, S., and Xie, Y.: Enhanced sulfate formation during China's severe winter haze episode in January 2013 missing from current models, Journal of Geophysical Research: Atmospheres, 119, 10,425-410,440, 10.1002/2013JD021426, 2014.

Zhao, Y., Chen, Z., Shen, X., and Zhang, X.: Kinetics and Mechanisms of Heterogeneous Reaction of Gaseous Hydrogen Peroxide on Mineral Oxide Particles, Environmental Science & Technology, 45, 3317-3324, 10.1021/es104107c, 2011.

Zheng, B., Zhang, Q., Zhang, Y., He, K. B., Wang, K., Zheng, G. J., Duan, F. K., Ma, Y. L., and Kimoto, T.: Heterogeneous chemistry: a mechanism missing in current models to explain secondary inorganic aerosol formation during the January 2013 haze episode in North China, Atmos. Chem. Phys., 15, 2031-2049, 10.5194/acp-15-2031-2015, 2015.

---

## Author Comment (AC2) · 9 Jul 2019

**Response to referee #2**

*The manuscript "Relative importance of gas uptake on aerosol and ground surfaces characterized by equivalent uptake coefficients" presented a theoretical approach to characterize the relative importance of uptake of trace gases on aerosols versus on ground. The authors proposed a new parameter "equivalent uptake coefficient" ($\gamma_{eqv}$) at which the flux of gas uptake on aerosols is equal to that on ground and derived $\gamma_{eqv}$ under various environment (vertical velocity and particle surface concentration). By comparing $\gamma_{eqv}$ with the effective uptake coefficient of gases on aerosols ($\gamma_{eff}$) reviewed from literature, the authors assessed the relative importance of gas uptake on aerosols to dry deposition. It was found that under urban environment, gas uptake on all types of aerosols (mineral dust, sea salt, organic aerosol, and soot) is important, while in pristine Amazonia forest the contribution of uptake on aerosols to gas loss is minor. $N_2O_5$ uptake on all types aerosol, $HNO_3$ and $H_2O_2$ on mineral aerosols, $O_3$ on liquid organic aerosol, $NO_2$, $SO_2$ and $HNO_3$ on sea salt aerosol are as important as dry deposition. The author also pointed out that $H_2O_2$ uptake on various aerosols need further laboratory studies and to be evaluated. The approach presented is a novel and convenient way to compare the relative importance of uptake of gases on aerosols with dry deposit. This manuscript is well written and easy to follow. And the discussion is well balanced. I have only a few minor comments, mainly to clarify some discussion. I recommend the direct publication of this manuscript on ACP after these minor comments are fixed.*

**Response:** We thank the positive and constructive comments given by the referee #2, which are very helpful to improve the manuscript. Our response to each specific comment is presented below.

*1. Pg. 4 line 18, a typical value of $\omega$ of 300 m s$^{-1}$ is used. I understand this can simplify the equation and $\gamma_{eqv}$, since different gases have slightly different mean velocity, especially in order to get a clear picture as shown in Fig. 2. Are the $\gamma_{eqv}$ values in Fig. 3-5 also calculated in this way? It might be helpful to briefly mention the influence of this simplification in the discussion part "Sect. 4.3".*

**Response:** We thank the referee's comments. We applied the same formula in Fig. 2 and Fig. 3-5, i.e., the typical mean thermal velocity of 300 m s$^{-1}$ was also used for Fig. 3-5. The biases due to this simplification are within 20% for calculations of $\gamma_{eqv}$ for $O_3$, $NO_2$, $SO_2$ and $HNO_3$, and within 30% for $H_2O_2$ and $N_2O_5$. We add more discussion on this simplification in the revised manuscript as follows:

"We use a unified thermal velocity (300 m s$^{-1}$) for all gases, which will introduce positive biases of +4% ~ +30% for $O_3$, $NO_2$, $SO_2$, $HNO_3$, $H_2O_2$, and a negative bias of -24% for $N_2O_5$ in calculations of $\gamma_{eqv}$ at the same temperature"

*2. Pg. 10 line 11, I am curious why the authors mainly discussed the model schemes in the studies Liao and Seinfeld (2005) and Wang K et al. (2012) among other model studies including heterogeneous reactions.*

**Response:** We thank the referee's comments. The scheme of Liao and Seinfeld (2005) and Wang K et al. (2012) were taken as an example here considering the large impact/applications of this scheme within the community (e.g., Monks et al., 2009; Wang et al., 2014; Li et al., 2015; Zheng et al., 2015). We update the table in the revised manuscript by including the scheme of Zhu et al. (2010) which uses updated values recommended by IUPAC (International Union of Pure and Applied Chemistry). It should be addressed that we only provide examples of model schemes here to give an overall implication for modelers, rather than to give a complete overview of the parameterizations of uptake coefficients covering all modeling studies, which is out of the scope of this paper. To avoid misunderstanding, we update the table, revise the caption to "Examples of aerosol uptake coefficients used in atmospheric models", add more illustrations in the footnote and main text, and move Table 2 to the supplement.

*3. Pg. 11 line 24, "…Sect. 3.5.1…", I guess that the authors meant "4.1.1". Also check line 26.*

**Response:** Thanks for the careful reading and help. We correct it in the revised manuscript.

*4. Pg. 13 line 27-Pg. 14 line 5, it might be helpful to also mention that the variability of aerosol surface concentration under each environment could also contribute to the variability of $\gamma_{eqv}$.*

**Response:** We agree with the referee's comments that the variability of aerosol surface concentration can contribute to the variability of $\gamma_{eqv}$. We have included the following statement to emphasize it in the revised manuscript:

"In addition, the variability of aerosol surface area under each environment can also contribute to the variability of $\gamma_{eqv}$."

*5. Pg. 14 line 25, it seems that one leading sentence is missing before "(a)…". Please double check.*

**Response:** We appreciate the referee's careful reading and help. We add a leading sentence before the statements (a)~(c).

"There are several indications from this work of processes that should be addressed in future measurements and model implementations:"

*6. Pg. 14 line 20, "…$HNO_3$ and $H_2O_2$ on mineral…", according to Fig. 2 should $SO_2$ be also listed here?*

**Response:** As shown in Fig. 4, there are more than three orders of magnitude of variances in $\gamma_{eff}$ for $SO_2$. $\gamma_{eff}$ of mineral dust falls in the range of $\gamma_{eqv}$ under high aerosol loadings or high mixing heights. The wide range of $\gamma_{eff}$ for mineral dust ($1.5 \times 10^{-8}$ to $6.3 \times 10^{-4}$) is a big

challenge regarding its application in models. Considering this large variations, we add $SO_2$ uptake on mineral dust as one of the important processes compared to dry deposition, and further discuss the potential uncertainty of $SO_2$ in item (c) of the "Conclusion" section.

*7. Pg. 38 line 6, "…the purple bar…" should be "blue bar".*

**Response:** Revised.

**References**

Li, X., Zhang, Q., Zhang, Y., Zheng, B., Wang, K., Chen, Y., Wallington, T. J., Han, W., Shen, W., Zhang, X., and He, K.: Source contributions of urban PM2.5 in the Beijing–Tianjin–Hebei region: Changes between 2006 and 2013 and relative impacts of emissions and meteorology, Atmospheric Environment, 123, 229-239, https://doi.org/10.1016/j.atmosenv.2015.10.048, 2015.

Monks, P. S., Granier, C., Fuzzi, S., Stohl, A., Williams, M. L., Akimoto, H., Amann, M., Baklanov, A., Baltensperger, U., Bey, I., Blake, N., Blake, R. S., Carslaw, K., Cooper, O. R., Dentener, F., Fowler, D., Fragkou, E., Frost, G. J., Generoso, S., Ginoux, P., Grewe, V., Guenther, A., Hansson, H. C., Henne, S., Hjorth, J., Hofzumahaus, A., Huntrieser, H., Isaksen, I. S. A., Jenkin, M. E., Kaiser, J., Kanakidou, M., Klimont, Z., Kulmala, M., Laj, P., Lawrence, M. G., Lee, J. D., Liousse, C., Maione, M., McFiggans, G., Metzger, A., Mieville, A., Moussiopoulos, N., Orlando, J. J., O'Dowd, C. D., Palmer, P. I., Parrish, D. D., Petzold, A., Platt, U., Pöschl, U., Prévôt, A. S. H., Reeves, C. E., Reimann, S., Rudich, Y., Sellegri, K., Steinbrecher, R., Simpson, D., ten Brink, H., Theloke, J., van der Werf, G. R., Vautard, R., Vestreng, V., Vlachokostas, C., and von Glasow, R.: Atmospheric composition change – global and regional air quality, Atmospheric Environment, 43, 5268-5350, https://doi.org/10.1016/j.atmosenv.2009.08.021, 2009.

Wang, Y., Zhang, Q., Jiang, J., Zhou, W., Wang, B., He, K., Duan, F., Zhang, Q., Philip, S., and Xie, Y.: Enhanced sulfate formation during China's severe winter haze episode in January 2013 missing from current models, Journal of Geophysical Research: Atmospheres, 119, 10,425-410,440, 10.1002/2013JD021426, 2014.

Zheng, B., Zhang, Q., Zhang, Y., He, K. B., Wang, K., Zheng, G. J., Duan, F. K., Ma, Y. L., and Kimoto, T.: Heterogeneous chemistry: a mechanism missing in current models to explain secondary inorganic aerosol formation during the January 2013 haze episode in North China, Atmos. Chem. Phys., 15, 2031-2049, 10.5194/acp-15-2031-2015, 2015.

Zhu, S., Butler, T., Sander, R., Ma, J., and Lawrence, M. G.: Impact of dust on tropospheric chemistry over polluted regions: a case study of the Beijing megacity, Atmos. Chem. Phys., 10, 3855-3873, 10.5194/acp-10-3855-2010, 2010.

---

## Author Response (AR1)

**Response to referee #1**

*In this work, Li et al. proposed "equivalent uptake coefficient", used this term to compare the relative importance of gas uptake onto aerosol surface versus group surface, and concluded that some uptake processes onto aerosol particles can be very important. The methodology is novel, and the results can be interesting for the atmospheric chemistry community. The manuscript can be accepted after the following comments are addressed.*

**Response:** We thank the positive and constructive comments given by the referee #1, which are very helpful to improve the manuscript. Our response to each specific comment is presented below.

Detailed Comments and Responses:

*1. P41, Table 1: There are some experimental studies (by Joel A Thornton, Jon Abbatt, Tim Bertram, and likely other) which explored the effect of organics on $N_2O_5$ uptake. In addition, there may be more studies on $H_2O_2$ uptake. Please check the IUPAC evaluation online as well as relevant literature.*

**Response:** We thank the referee's comments. In the last version, we have already included the following experimental studies suggested by the referee in Table A.3 regarding $N_2O_5$ uptake on organics, i.e., Thornton et al. (2003), Griffiths et al. (2009), and Badger et al. (2006). In the revised manuscript, we have tried to complete the list by including Folkers et al. (2003), Gross et al. (2009) and Anttila et al. (2006) as in the new Table A.3.

For $H_2O_2$, we checked the IUPAC evaluation online data and related literature. We now add one more measurement of $H_2O_2$ uptake on mineral dust (Zhao et al., 2011) in the revised manuscript. We are still unable to find more laboratory measurements of $H_2O_2$ uptake on aerosols other than mineral dust, thus more measurements are needed in the future.

*2. P42, Table 2: I am not sure why the work by Wang et al. (2012) is used a representative example here. In fact, the uptake coefficients used by Wang et al. are far from being updated, and they mainly used uptake coefficients adopted by two modeling studies almost 20 years ago (Dentener et al., 1996; Zhang and Carmichael, 1999). For mineral dust in specific, the uptake coefficients used by Zhu et al. 2010 were updated values recommended by IUPAC. In addition, some of the studies which are cited as the sources of uptake coefficients measured by laboratory work are in fact pure modeling work, such as Bauer et al. (2004), Dentener et al. (1996), and so on. The author may consider updating this table.*

**Response:** We thank the referee's comments. The scheme of Wang K et al. (2012) is taken as an example considering the large impact/applications of this scheme within the community (e.g., Wang et al., 2014; Li et al., 2015; Zheng et al., 2015). We update the table in the revised manuscript by including the scheme of Zhu et al. (2010) which uses updated values recommended by IUPAC. It should be addressed that we only provide examples of model

schemes that have considered the heterogeneous reactions to give an overall implication for modelers, rather than to give a complete overview of the parameterizations of uptake coefficients covering all modeling studies, which is out of the scope of this paper. To avoid misunderstanding, we revise the caption to "Examples of aerosol uptake coefficients used in atmospheric models", add more illustrations in the footnote and main text, and move Table 2 to the supplement.

Uptake coefficients with sources listed as Dentener et al. (1996), Bauer et al. (2004), Song and Carmichael (2001) in the table are from model parameterization, and the specific laboratory measurements are not found in the literature. We have already noted this issue in the last version (see footnote [b]).

*3. P3, L23-25: HO$_2$ uptake can be very important for tropospheric chemistry (George et al., 2013; Mao et al., 2013; Taketani et al., 2008; Thornton et al., 2008). Is there a reason why HO$_2$ has not been discussed in this paper?*

**Response:** We agree that HO$_2$ uptake on aerosol is important for atmospheric chemistry. The motivation of our work is to compare the fluxes of dry deposition and aerosol uptake, which is difficult for radicals like HO$_2$ because the required parameters to calculate dry deposition are not available from current literature. The reason could be that the other pathways are too fast compared to the deposition of HO$_2$.

*4. P2, L29-30: Very recently I reviewed heterogeneous reactions of mineral dust (Tang et al., 2017). Should this paper be cited here?*

**Response:** We have included this review paper in the revised manuscript along with other original research articles.

*5. P3, L12-15: Another convenient way to assess the relative importance of aerosol uptake and dry deposition is to calculate their lifetimes with respect to individual processes, as discussed by Tang et al. (2017).*

**Response:** We agree with the referee that comparison of lifetimes between aerosol uptake and dry deposition is another feasible method, as discussed in Tang et al. (2017). The method proposed in our study share the same basic formulations with Tang et al. (2017), and velocities can be easily converted to lifetime.

*6. P4, L24-26: This sentence is not easy to follow. I assume that the authors wanted to state that for smaller particles, gas phase diffusion would not be a limiting step and thus can be neglected. Please consider rephrasing it, and refer to Tang et al. (2014) for a comprehensive discussion on the role of gas phase diffusion.*

Response: We thank the referee's comments. We have referred to the original formation of Jacob 2000 as follows:

"For atmospheric aerosols with a diameter of ~0.2 μm or smaller, the related gaseous uptake tends to be limited by the free molecular collision rate (uptake rate $\rightarrow \omega\alpha A[X_g]/4$) (Jacob, 2000). Thus in the following analyses, we mainly focus on the discussion of $\gamma_{eff}$, and neglect the diffusion resistance in the gas phase."

*7. P12-13: In a paper published in 2017 (Tang et al., 2017), I provided a comprehensive and in depth discussion on the two factors the authors mentioned in Section 4.2, and would like to refer the authors to take a look at that paper.*

**Response:** Thanks for the referee's comments. We refer to this work in our discussion as follows:

"More than three orders of magnitudes of differences are derived by whether to consider the pores within the microstructure of solid aerosol surface or not (see Table A.1). Using the same method to calculate the available surface area may reconcile these differences (Tang et al., 2017)."

*8. P14, L19-21 as well as related content in the abstract It is proposed that the following four groups of gas uptake onto aerosols can be important: 1) $N_2O_5$ on all types of aerosols, 2) $HNO_3$ and $H_2O_2$ on mineral dust, 3) $O_3$ on liquid organic aerosols; and 4) $NO_2$, $SO_2$, $HNO_3$ on sea salt aerosols. The four groups have some overlaps and not easy to follow. I would suggest re organizing them according to either types of gases or types of aerosol s*

**Response:** This is re-organized to present the most intensive summary of our conclusion.

We will add the following table to better illustrate it. According to Table 2, there is no overlapped information and four bullets are already the minimum number of vectors from this matrix.

**Table 2. Gas uptake processes that are potentially important compared to dry deposition across various environments (marked with √).**

| Gases | Mineral dust | Soot | Organic aerosol-solid | Organic aerosol-liquid | Sea salt aerosol |
|-------|--------------|------|-----------------------|------------------------|------------------|
| $O_3$ | | | | √ | |
| $NO_2$ | | | | | √ |
| $SO_2$ | √ | | | | √ |
| $N_2O_5$ | √ | √ | √ | √ | √ |
| $HNO_3$ | √ | | | | √ |
| $H_2O_2$ | √ | | | | |

|---|---|---|---|---|---|---|---|
| $O_3$ | Mineral dust | $1.0\times10^{-5}$ | Michel et al., 2002, 2003 | $2.7\times10^{-5}$ | IUPAC[e] | $5.0\times10^{-5} \sim 1.0\times10^{-4}$ | Dentener et al., 1996[b]; Zhang and Carmichael, 1999[b] |
| $NO_2$ | Mineral dust | | | $2.1\times10^{-6}$ | IUPAC | $4.4\times10^{-5} \sim 2.0\times10^{-4}$ | Underwood et al., 2001 |
| | Wet aerosol | $1.0\times10^{-4}$ | Jacob, 2000 | | | | |
| $SO_2$ | Mineral dust | $3.0\times10^{-4}$(RH<50%), 0.1(RH≥50%) | Dentener et al., 1996 | $3.0\times10^{-5}$ | IUPAC | $1.0\times10^{-4} \sim 2.6\times10^{-4}$ | Zhang and Carmichael, 1999[b] |
| | Sea salt aerosol | $5.0\times10^{-3}$(RH<50%), $5.0\times10^{-2}$ (RH≥50%) | Song and Carmichael, 2001[b] | | | | |
| $N_2O_5$ | Mineral dust | See footnote[c] | Bauer et al., 2004[b] | $3.0\times10^{-2}$ | Seisel et al., 2005; Wagner et al., 2008; Karagulian et al., 2006 | $1.0\times10^{-3} \sim 0.1$ | Dentener et al., 1996; DeMore et al., 1997 |
| | Organic carbon | $5.2\times10^{-4}\times$ RH(RH<50%), 0.03(RH≥50%) | Thornton et al., 2003 | | | | |
| | Sea salt aerosol | $5 \times10^{-3}$ (RH<50%), 0.03 (RH≥50%) | Atkinson et al., 2004 | | | | |

| | Sulfate/nitrate/ammonium | See footnote [d] | Kane et al., 2001, Hallquist et al., 2003 | | | | |
|---|---|---|---|---|---|---|---|
| $HNO_3$ | Mineral dust | 0.1 | Hanisch and Crowley, 2001 | 0.17 | IUPAC | $1.1\times10^{-3} \sim 0.2$ | Dentener et al., 1996; DeMore et al., 1997; Underwood et al., 2001 |
| $H_2O_2$ | Mineral dust | | | $2.0\times10^{-3}$ | De Reus et al., 2005 | $1.0\times10^{-4} \sim 2.0\times10^{-3}$ | Dentener et al., 1996 |

[a] Here we present two parameterization schemes as examples: the full scheme of Liao and Seinfeld (2005), the scheme for mineral dust of Zhu et al. (2010) and Wang et al. (2012). The original references of the measurements regarding the uptake coefficients are listed. It should be addressed that these schemes are only examples of modelling studies.

[b] Model parameterization. The specific references to laboratory measurements for uptake coefficients are not found.

[c] $\gamma = 4.25\times10^{-4}\times RH - 9.75\times10^{-3}$

[d] $\gamma = 10^{\beta(T)}\times(C_1 + C_2\times RH + C_3\times RH^2 + C_4\times RH^3)$

$\beta(T) = -4\times10^{-2}\times(T - 294), T \geq 282K$

$\beta(T) = 0.48, T < 282K$

$C_1 = 2.79\times10^{-4}$; $C_2 = 1.30\times10^{-4}$; $C_3 = -3.43\times10^{-6}$; $C_4 = 7.52\times10^{-8}$

[e] IUPAC: International Union of Pure and Applied Chemistry, available at http://iupac.pole-ether.fr/

**Reference**

[revised manuscript text omitted]